# Discrete LAT condensates encode antigen information from single pMHC:TCR binding events

Darren B. McAffee [1], Mark K. O'Dair [1], Jenny J. Lin[1], Shalini T. Low-Nam[1], Kiera B. Wilhelm [1], Sungi Kim[1], Shumpei Morita[1] & Jay T. Groves [1,2] ✉

LAT assembly into a two-dimensional protein condensate is a prominent feature of antigen discrimination by T cells. Here, we use single-molecule imaging techniques to resolve the spatial position and temporal duration of each pMHC:TCR molecular binding event while simultaneously monitoring LAT condensation at the membrane. An individual binding event is sufficient to trigger a LAT condensate, which is self-limiting, and neither its size nor lifetime is correlated with the duration of the originating pMHC:TCR binding event. Only the probability of the LAT condensate forming is related to the pMHC:TCR binding dwell time. LAT condenses abruptly, but after an extended delay from the originating binding event. A LAT mutation that facilitates phosphorylation at the PLC-γ1 recruitment site shortens the delay time to LAT condensation and alters T cell antigen specificity. These results identify a function for the LAT protein condensation phase transition in setting antigen discrimination thresholds in T cells.

A healthy adaptive immune response depends on the ability of T cells to discriminate between agonist peptide major histocompatibility complex (pMHC) ligands and the vastly more abundant self pMHC. Antigen discrimination is based on the binding kinetics of pMHC to T cell receptor (TCR), especially the kinetic off-rate (or, equivalently, the mean binding dwell time)[1–3]. Agonist and self-ligands may differ only slightly in their binding kinetics, requiring a highly precise discrimination mechanism[4–9]. Furthermore, T cells can accurately discriminate pMHC ligands based on only a small number (tens) of individual molecular binding events[10–14]. A quantitative mapping of the antigen discrimination function for an individual TCR has yet to be directly measured, and estimates based on indirect bulk parameters span a wide range[15–19]. Precise tuning of the T cell signaling system is crucial for a proper immune response. Mutations in early signaling proteins that increase TCR sensitivity are associated with autoimmune diseases, while mutations that decrease TCR sensitivity are associated with immunodeficiency[20].

Binding of pMHC to TCR initiates a signaling process, the first few steps of which rely on sustained engagement of the pMHC:TCR complex[2,17,21]. This establishes a kinetic proofreading mechanism, in which only sufficiently long dwelling pMHC:TCR binding events successfully trigger the full downstream signaling response[8,15,22]. Following the initial formation of the pMHC:TCR complex, immune receptor tyrosine-based activation motifs (ITAMs) on the TCR CD3 chains are phosphorylated by the Src family kinase, Lck, creating docking sites for the Syk kinase, ZAP-70[17,23]. ZAP-70 arrives to the TCR in an initially autoinhibited state, but is activated after phosphorylation by Lck, and subsequently phosphorylates substrates including linker for activation of T cell (LAT)[24–27]. A distinctive feature of the TCR signaling mechanism is that LAT is a poor substrate for most kinases[28], and thus relies primarily on ZAP-70. This requisite series of kinase activation steps leading up to ZAP-70 activation at the TCR forms the first stage of kinetic antigen discrimination.

Under the control of phosphorylation by ZAP-70, LAT scaffolds a signaling hub downstream of TCR from which both $Ca^{2+}$ and MAPK signal pathways branch[29–32]. LAT is an intrinsically disordered protein anchored to the membrane via a single transmembrane domain. LAT contains 9 tyrosine phosphorylation sites, at least 4 of which (Y136,

[1]Department of Chemistry, University of California, Berkeley, Berkeley, CA 94720, USA. [2]Institute for Digital Molecular Analytics and Science, Nanyang Technological University, 59 Nanyang Drive, Singapore 636921, Singapore. ✉e-mail: jtgroves@lbl.gov

Y175, Y195, Y235 in mouse LAT) are utilized in T cell signaling. The Src-homology 2 (SH2) domain-containing adapter protein Grb2 binds to phosphorylated LAT residues and subsequently recruits the Ras guanine nucleotide exchange factor (GEF), Son of Sevenless (SOS), which activates Ras and the MAPK pathway. Phosphorylation at tyrosine 136 on mouse LAT (Y132 on human LAT) generates a binding site selective for PLC-γ1 recruitment[33], which further interacts with GADS and SLP76, and ultimately activates $Ca^{2+}$ signaling[29]. MAPK and $Ca^{2+}$ signaling in T cells leads to ERK and NFAT nuclear translocation, respectively, and ultimately controls IL-2 production, T cell differentiation, and other effector functions[34]. The high multivalency of LAT phosphotyrosine sites, along with several crosslinking interactions among various adapter proteins, enables extended clustering into an elaborate signaling complex[35–37]. More recent studies of LAT, Grb2, and SOS reconstituted on supported membranes have reported a two-dimensional protein condensation phase transition governed by tyrosine phosphorylation[38,39]. Formation of this condensate was further discovered to facilitate release of autoinhibition in SOS, thus enabling Ras activation, and possibly providing a signal-gating function in T cells[40].

Here we examine the signaling process from initial binding of individual pMHC:TCR molecular complexes to the associated LAT condensation. T cells are highly sensitive to antigen and can robustly activate at agonist pMHC densities so low that pMHC ligands are spaced microns apart and can be readily resolved by single-molecule imaging[11,12,41]. We utilize the hybrid live cell-supported membrane system, in which a supported membrane functionalized with pMHC and intercellular adhesion molecule-1 (ICAM-1) forms a surrogate antigen-presenting cell (APC) surface for interactions with primary mouse T cells. This experimental platform has been used extensively in the context of the immunological synapse[42–49], and provides an optimal configuration for single-molecule imaging by total internal reflection fluorescence (TIRF) microscopy[13,45,50]. Individual pMHC molecules in the supported membrane can be tracked with high spatial (50 nm) and temporal (20 ms) resolution. In these experiments, free pMHC molecules exhibit simple two-dimensional Brownian motion in the membrane and nonspecifically immobilized pMHC represent a negligibly small fraction (<1%). When pMHC binds TCR on an opposed T cell, however, its motion changes dramatically, allowing clear distinction of pMHC:TCR complexes from free pMHC ligand[12,13]. Here, we track the formation, duration, and movement of individual pMHC:TCR complexes while simultaneously monitoring LAT condensation (as well as localization of other signaling molecules) in response to each pMHC:TCR binding event. Unique to this experimental strategy is the ability to map the LAT condensation and other signaling responses to the specific molecular binding event from which they originated.

The observations imply that a single pMHC:TCR binding event is sufficient to trigger formation of a two-dimensional condensate on the membrane containing hundreds of LAT molecules. LAT condensation occurs abruptly, but after an extended delay from the originating binding event, exhibiting signatures of a phase transition. The resulting LAT condensates are self-limiting, and neither their size nor their lifetime is correlated with the duration of the originating pMHC:TCR binding event. Only the probability of forming a LAT condensate is related to the pMHC:TCR binding dwell time. We report quantitative measurements of this probability distribution, providing experimental mapping of a single TCR antigen discrimination function. These results implicate extended kinetic discrimination of ligand, into the tens of seconds binding dwell times. This long timescale discrimination represents an additional layer of kinetic proofreading, extending beyond the TCR itself, which is provided by the LAT condensation phase transition. We further observe that a LAT mutation (G135D in mouse, G131D in human), which enhances the kinetics of ZAP-70 phosphorylation at Y136 on LAT (the PLC-γ1 recruitment site)[51], decreases the delay time to LAT condensation and alters T cell antigen

specificity. Whole-cell activation (measured by NFAT translocation) correlates with the number of LAT condensates formed after exposure to pMHC, suggesting that LAT condensates represent quanta of information in the T cell signaling pathway. We suggest that the LAT condensation phase transition has evolved in TCR signaling as a means to achieve the amplification and noise suppression necessary for the single-molecule ligand sensitivity exhibited by T cells.

## Results

### Single pMHC:TCR binding events trigger discrete LAT condensates

We characterize LAT condensation in response to pMHC:TCR binding events using a hybrid live cell-supported membrane experimental platform[44,52–55] (Fig. 1a). The supported membrane, consisting primarily of DOPC lipids, is functionalized with pMHC (0–100 molecules μm$^{-2}$) and ICAM-1 (300–600 molecules μm$^{-2}$), both of which are linked to the membrane via His-tag-protein:Ni-chelating-lipid interactions[56,57]. We utilize multicolor TIRF imaging to track LAT condensation at the membrane in response to T cell engagement with the pMHC and ICAM-1 functionalized supported bilayers. Experiments are generally performed using primary T cells harvested from mice with transgenic TCR(AND)[58]. Fluorescent fusion proteins of interest were expressed using the PLAT-E retroviral platform[59]; for experiments requiring expression of two different fluorescent fusion proteins we utilized a self-cleaving P2A peptide[60]. Reflection interference contrast microscopy (RICM) was used as a way of monitoring the cell-supported membrane contact independently from the fluorescence channels[61].

TIRF microscopy of the live cell-supported membrane interface provides robust imaging capabilities all the way down to the single molecule level[50,62,63]. At low pMHC densities (0.1–0.4 molecules μm$^{-2}$) individual pMHC molecules can be tracked undergoing free Brownian motion laterally in the membrane, with diffusion coefficients of $0.55 \pm 0.07$ μm$^{2}$s$^{-1}$ as determined from single-molecule step size distributions. When pMHC binds to a TCR on a T cell, its motion changes dramatically to conform with the much slower movement of the TCR (Supplementary Movie 1, Supplementary Fig. 1a)[12,13,64,65]. Under long exposure times (500 ms) and low excitation powers (0.4 mW), individual pMHC:TCR complexes appear as well-defined spots whereas free pMHC is moving too quickly to be clearly imaged, and appears as a diffuse background (Supplementary Fig. 1b). This imaging method offers selective tracking of pMHC:TCR complexes in living cells with time resolution of about one second and spatial resolution of ≈300 nm[12,13]. Trajectories for each pMHC:TCR complex are tracked to measure the specific pMHC:TCR binding dwell time as well as their spatial movement within the T cell membrane.

Representative TIRF images of primary T cells expressing LAT-eGFP interacting with stimulatory bilayers containing various densities of agonist pMHC are illustrated in Fig. 1b (see also Supplementary Movie 2). LAT condensation is readily visible as localized increases in LAT density and corresponding time traces of the overall extent of LAT condensation are plotted in Fig. 1c. At high agonist pMHC densities of 10–40 molecules μm$^{-2}$, LAT condensation is observed throughout the cell-supported membrane interface within seconds of cell landing. This is consistent with numerous reports of LAT clustering and condensation in T cells interacting with APCs displaying high agonist pMHC density[42,66–68] as well as T cells interacting with activating anti-TCR antibody coated surfaces[69,70]. At these high antigen densities, LAT phosphorylation and subsequent condensation is driven by multiple TCR activation events, which are also intrinsically unsynchronized in time and are superimposed in the images. The observed LAT condensation is a conglomerate of correspondingly unsynchronized individual nucleation events (Supplementary Fig. 1c), which cannot be directly mapped to the specific pMHC:TCR binding events that triggered them; and much information is lost. At lower agonist pMHC densities (Fig. 1b, middle panel, also see Supplementary Movie 3), the

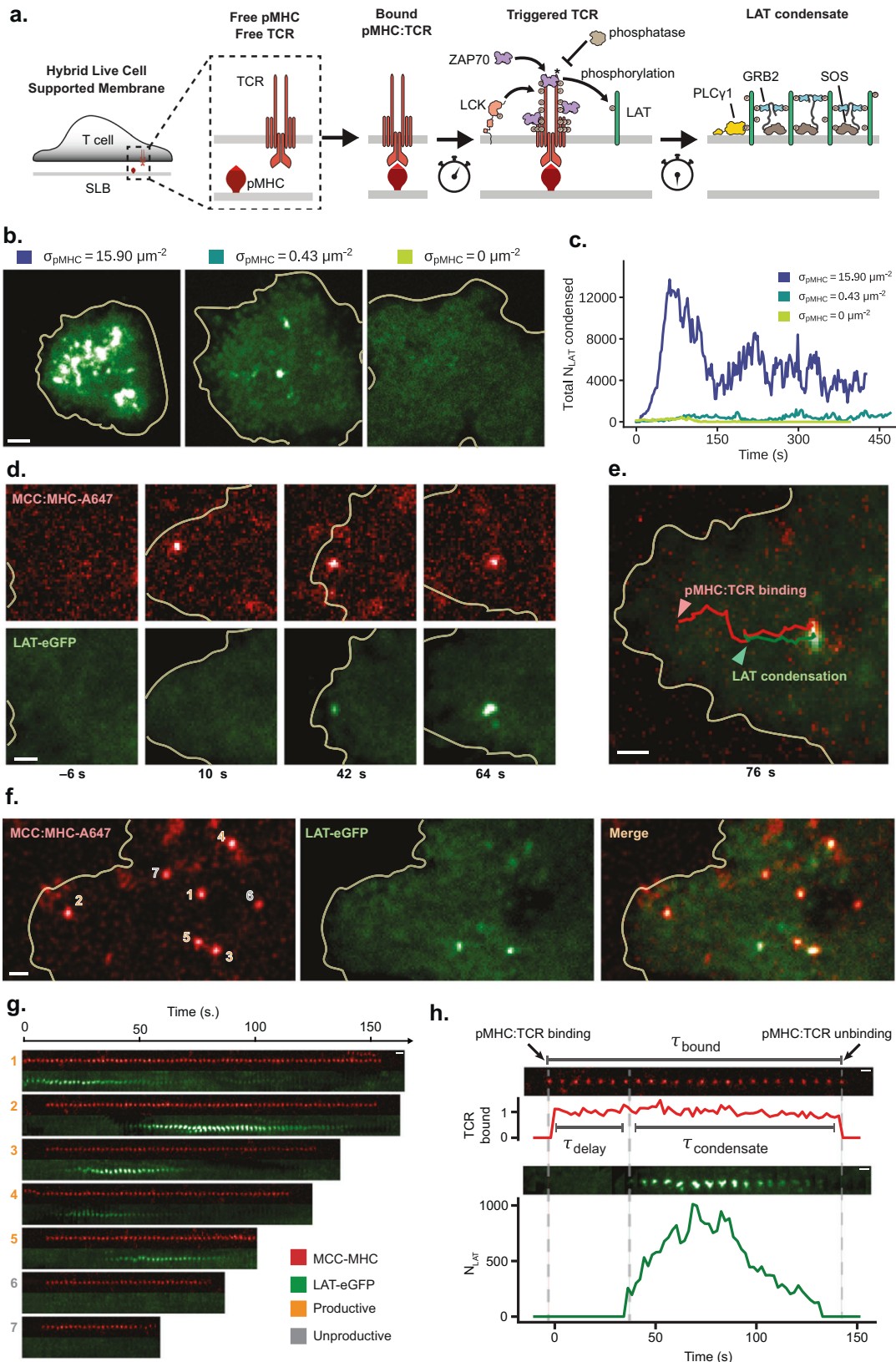

overall amount of LAT condensation is lower, however, similarly high local densities of LAT are still observed to form stochastically and widely distributed throughout the interface. Although the total amount of LAT condensation remains low, and appears to develop on a much longer timescale, T cells still activate robustly at these pMHC densities[10–12,41]. In control experiments with zero pMHC in the

membrane (Fig. 1b, right panel), very few LAT condensates are observed and T cells do not activate. The different timescales observed for LAT condensation at low and high antigen are in good agreement when the superposition of intrinsically stochastic individual binding events is taken into consideration (see "Simulation of whole cell inte-grated LAT signal" in Methods and Supplementary Figs. 1c–e).

**Fig. 1 | Single pMHC:TCR binding events trigger discrete LAT condensates.** The representative microscopy images in this figure were reproducible across 8 separate experiments/mice. **a** Schematic of T cell signaling that occurs downstream of pMHC:TCR binding after the plasma membrane of the T cell interfaces with the supported lipid bilayer (SLB). **b** Representative TIRF images of primary mouse CD4+ T cells expressing AND TCR and LAT-eGFP deposited onto SLBs with varying densities of MCC(Atto647) pMHC. **c** The amount of condensed LAT through time for cells in panel **b**. and was calculated by dividing the intensity of the total condensed area (as determined by simple thresholding) by the intensity of single LAT-eGFP molecules, see "Methods" for more detail. **d** Time series images of a spatially isolated pMHC:TCR binding event that produces a highly localized LAT condensate within 50 nm at onset of condensation. Time stamps are relative to the moment of pMHC:TCR binding. **e** Overlay of MCC(Atto647) pMHC and LAT-eGFP channels showing the co-localized trajectories of the pMHC:TCR binding event and its associated LAT condensate, as well as their centripetal motion towards the center of the cell. **f** A wider view of the T cell shows a constellation of binding events spread out in space. Separate binding events are enumerated. **g** Temporal sequence of images for the binding events in panel **f**. Binding events visible in the first frame of the data acquisition are not considered for subsequent analyses. Some binding events are productive (numbered in gold), while others fail to produce a localized LAT condensate (numbered in gray). **h** Intensity traces of productive binding events have several quantities—the number of condensed LAT produced ($N_{LAT}$), the delay time until LAT condensation ($\tau_{delay}$), as well as the lifetime of the LAT condensate ($\tau_{condensate}$). The scale bar in **b** is 2 µm, while all other scale bars are 1 µm. Related data are in Supplementary Fig. 1. Source data are provided as a Source data file.

At low pMHC densities (0.1–0.4 molecules µm$^{-2}$), individual pMHC:TCR binding events are spaced microns apart and can remain distinct for the entire lifetime of the bound complex. Individual pMHC:TCR binding events can nucleate LAT condensation. A time sequence of images illustrating the formation and movement of a single pMHC:TCR complex along with the corresponding LAT condensate it nucleated are illustrated in Fig. 1d (see also Supplementary Movie 4). LAT condensation begins within tens of nanometers of the originating pMHC:TCR complex and generally remains within a 100–300 nm neighborhood of the complex as both undergo co-localized transport towards the center of the cell (Fig. 1e and Supplementary Fig. 1f). This retrograde transport of single pMHC:TCR complexes along with their associated LAT condensates is very similar to the retrograde transport of TCR clusters that occurs at high pMHC density[64,65,71,72]. LAT condensates are observed to nucleate throughout the cell-supported membrane interface, with a slight preference for more peripheral positions (Supplementary Fig. 1g). The imaging methods we use here are also sensitive enough to distinguish rarely formed pMHC dimers (Supplementary Fig. 1h) from the predominant monomer pMHC:TCR complexes; we focus on monomer pMHC:TCR binding events at low overall agonist pMHC densities in this study. Over the entire cell interface, multiple such pMHC:TCR binding events and associated LAT condensates can be observed (Fig. 1f; Supplementary Movie 5). The LAT condensates form and ultimately dissipate, generally independently of each other; at any one moment in time there may only exist a few LAT condensates within the cell. A collection of image time sequences tracking pMHC:TCR binding along with the associated LAT condensate formation and subsequent dissipation for the seven binding events from the cell pictured in Fig. 1f are shown in Fig. 1g; note that two binding events failed to produce LAT condensates in this particular set.

Detailed analysis of a productive binding event is presented in Fig. 1h. The total pMHC:TCR binding dwell time ($\tau_{bound}$) is observed in the top (red) sequence of images. The sequence of images below (green) track the corresponding LAT condensate, with a calibrated trace plotting the number of LAT molecules in the condensate through time. LAT condensation occurs abruptly, but after a relatively long delay time (≈40 s in this trace; mean delay is $\langle\tau_{delay}\rangle \approx 23$ s). The LAT condensates are self-limiting with a mean lifetime of $\langle\tau_{condensate}\rangle \approx 30$ s; a similar limiting lifetime for LAT condensates has recently been estimated indirectly from mass spectrometry studies[73] on cell populations. In some cases (e.g. Fig. 1g and h), LAT condensates can dissipate prior to dissociation of the originating pMHC:TCR complex. Fluorescence recovery after photobleaching (FRAP) measurements on the LAT condensates (Supplementary Fig. 2a) indicate they are relatively dynamic, with individual LAT molecules turning over throughout the lifetime of the condensate (see "FRAP of LAT condensates" in Methods).

We measured the total number of LAT molecules in each pMHC:TCR-induced condensate using a combination of quantitative immunoblots (Supplementary Fig. 2b), fluorescence activated cell sorting (FACS), and quantitative fluorescence imaging (Fig. 2a, Supplementary Fig. 2c, Supplementary Movie 6, see "Quantitative fluorescence" in Methods). The observed LAT copy number in single condensates was measured to be a linear function of total LAT expression level (Fig. 2b). By extrapolating to endogenous LAT expression levels (zero expression of LAT-eGFP), we determine that $258 \pm 65$(SD) LAT molecules are contained within a physiological condensate produced by a single pMHC:TCR binding event. In Fig. 2c we plot data from an ensemble ($n = 100$) of pMHC:TCR binding events mapping the binding dwell time, which contains the physical information of antigen identity, to properties of the associated LAT condensate. Neither the size, nor the lifetime, of a LAT condensate is correlated with the duration of the originating pMHC:TCR binding event. As analyzed in greater detail below, only the probability of LAT condensation formation is related to pMHC:TCR binding dwell time.

The background distribution of LAT in primary T cell membranes is extremely uniform (as seen in Fig. 1b and d). Individual pMHC:TCR binding events produce distinctly resolved LAT condensates against this smooth background. Experimental examination shows that these pMHC:TCR-induced LAT condensates form from plasma membrane LAT and are distinct, in both composition and their association with agonist pMHC, from LAT vesicles arriving at the membrane from the cytosol (a small number of which can also be observed, see Supplementary Discussion 1 and Supplementary Figs. 2d–f). We also note that Jurkat T cells can exhibit greater variability and can have significant levels of LAT pre-clustering or pre-condensation in the absence of any TCR activation (Supplementary Fig. 2g, top), rendering them less suitable for these types of high-resolution single-molecule studies. Control experiments looking at primary human T cells indicated negligible preclustered LAT, while stimulation with anti-CD3 antibody confirmed clean LAT condensation events exist (Supplementary Fig. 2g, bottom). Thus, the above-mentioned issues are specific to the immortalized human Jurkat T cell line, and do not represent a difference between mouse and human primary T cells. In this study we focus on LAT condensates in whose origin can be directly associated to a pMHC:TCR binding event.

## pMHC:TCR binding dwell time sets probability of LAT condensation

A single pMHC:TCR binding event either produces a local LAT condensate or fails to do so (Supplementary Fig. 3). Although the size and lifetime of individual LAT condensates exhibit no correlation with the binding dwell time of the originating pMHC:TCR complex (Fig. 2c), the probability of producing a condensate is correlated. Here we define the success probability, $P_{LAT}(\tau_{MHC})$, to be the probability that a pMHC:TCR binding event of duration $\tau_{MHC}$ produces a LAT condensate at any point during its lifetime. We measure this success probability function for an individual TCR by examining an ensemble ($n = 1071$) of individual MCC pMHC:TCR binding events and mapping the corresponding LAT condensation outcome. Data from this ensemble is compiled in Fig. 3a, illustrating the pMHC:TCR binding dwell time (traced in red) as well as subsequent LAT condensation (traced in green), if it occurs. The delay time between initial pMHC:TCR

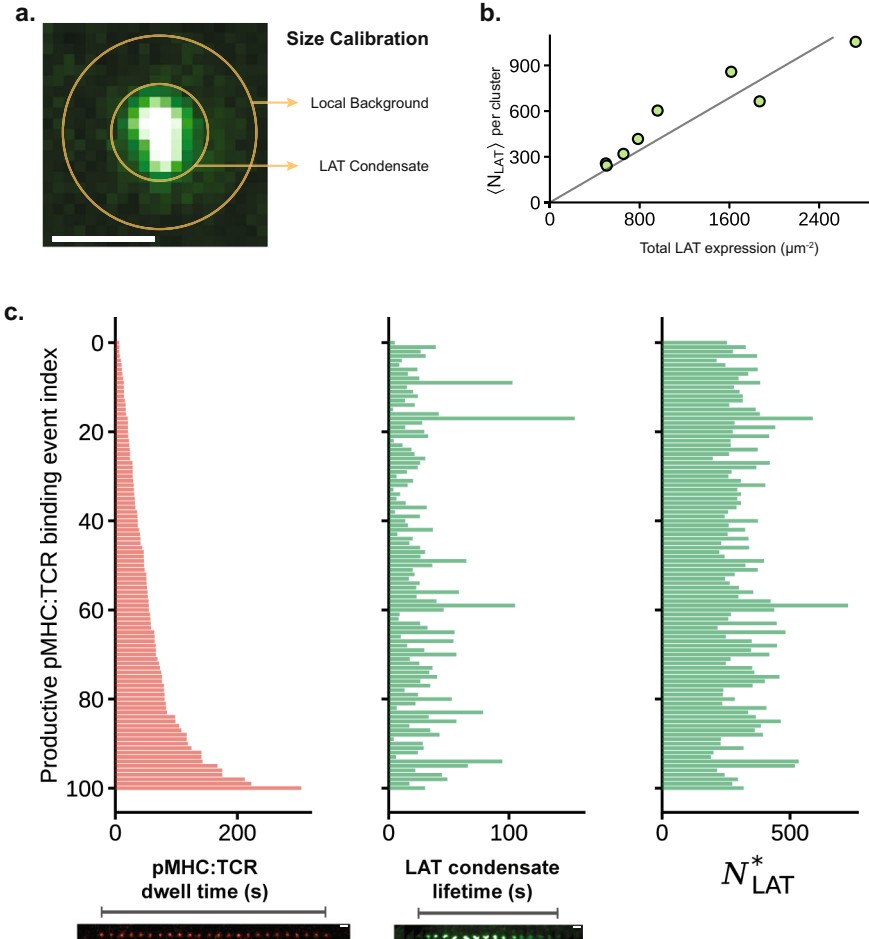

**Fig. 2 | Physical properties of the LAT condensate. a** An annulus around the LAT condensate is used to sample the local background. The intensity of the condensate is calculated as the midpoint of the net intensity and total intensity, since the extent of background LAT clustering is unknown. Scale bar is 1 μm. **b** Scatter plot of the average number of LAT within condensates, $\langle N_{LAT} \rangle$, as a function of the total LAT expression level (endogenous LAT + exogenous LAT-eGFP). Each point is a cell average over condensates observed within the first 5 min of cell landing. The fitted line has the equation $y = 0.43x$, which was constrained to have 0 intercept. All cells were from the same mouse. Cells from two other mice yielded similar fits, $y = 0.51x$ and $y = 0.48x$. **c** Three bar plots showing the lack of correlation between pMHC:TCR dwell time (left) with either the LAT condensate lifetime (middle) or the number of LAT within its associated condensate (right). Each red bar is a pMHC:TCR binding event, data is pooled from the 8 cells in panel **b**. The relative size of the LAT condensate is calculated as $RS = \frac{\text{Total Intensity} + \text{Net Intensity}}{2(1\,\mu m^2 \text{ of Background Intensity})}$. $N^*_{LAT}$ is the relative size (RS) of the condensate rescaled using an estimated physiological (zero exogenous) LAT density of $601 \pm 150(\text{SD})\,\mu m^{-2}$. This normalizes LAT cluster size to LAT expression level for cell-to-cell comparisons. Related data are in Supplementary Fig. 2. Source data are provided as a Source data file.

binding and LAT condensation ($\tau_{delay}$) is also visible in this data set, but here we focus exclusively on the binary success or failure of each event. The data is aggregated into bins based on pMHC:TCR dwell time, indicated by gray lines, and the resulting counts of all and successful binding events are plotted in Fig. 3b. A similar data set derived from the weaker T102S pMHC is plotted in Fig. 3c. The observed dwell time distributions from all binding events for both the strong (MCC) and moderate (T102S) agonist pMHC:TCR are exponentially distributed with mean dwell times of $23.8 \pm 2.4(\text{SE})$ and $9.1 \pm 2.8(\text{SE})$ s, respectively. These observed dwell times include events that end by unbinding as well as photobleaching; imaging parameters were adjusted to balance photobleaching effects and time resolution (see "Imaging" in Methods). These in situ dwell time measurements, when combined with photobleaching measurements, allow extraction of the true binding dwell times of $44 \pm 6$ (MCC) and $10.8 \pm 2$ (T102S) s (see "Dwell times" in Methods), which are in good agreement with bulk kinetic measurements for pMHC:TCR binding for MCC and T102S[74,75]. The success probability as a function of dwell time, $P_{LAT}(\tau_{MHC})$, based on the time bins plotted in Fig. 3a–c, is plotted in Fig. 3d (bottom). Since a binding event is considered successful if a LAT condensate forms at any point during its lifetime, this distribution is a calibrated

measure of probability; it is not impacted by the sampling efficiency for different binding dwell times (see "$P_{LAT}(\tau_{MHC})$ and delay time" in Methods). These data show that short dwelling binding events have near zero probability of producing a LAT condensate, but as binding dwell time increases, so does the probability of LAT condensation, with a $t_{\text{half}-\text{max}} = 24.0 \pm 1.3$ s. The longest dwelling events approach a maximum 0.25 probability of producing a LAT condensate. The similarity of the probability curves traced by both MCC and T102S pMHC reinforces the primacy of dwell time over peptide identity as the fundamental input to TCR triggering, at least between these two peptide antigens.

The data plotted in Fig. 3d represents measurement of a single TCR antigen discrimination function. It implies two mechanistic features of molecular antigen discrimination by TCR not readily discerned from bulk measurements. First, the long rise time after initial binding indicates that individual TCR are still sensitive to the binding dwell time of a pMHC ligand beyond 30 s from initial engagement. This is much longer than kinetic cut-offs for antigen discrimination observed at high antigen density (1–3 s)[18,19,76], and represents an additional layer of kinetic proofreading that becomes prominent when antigen counts are low. Although the measured probability of a pMHC:TCR complex

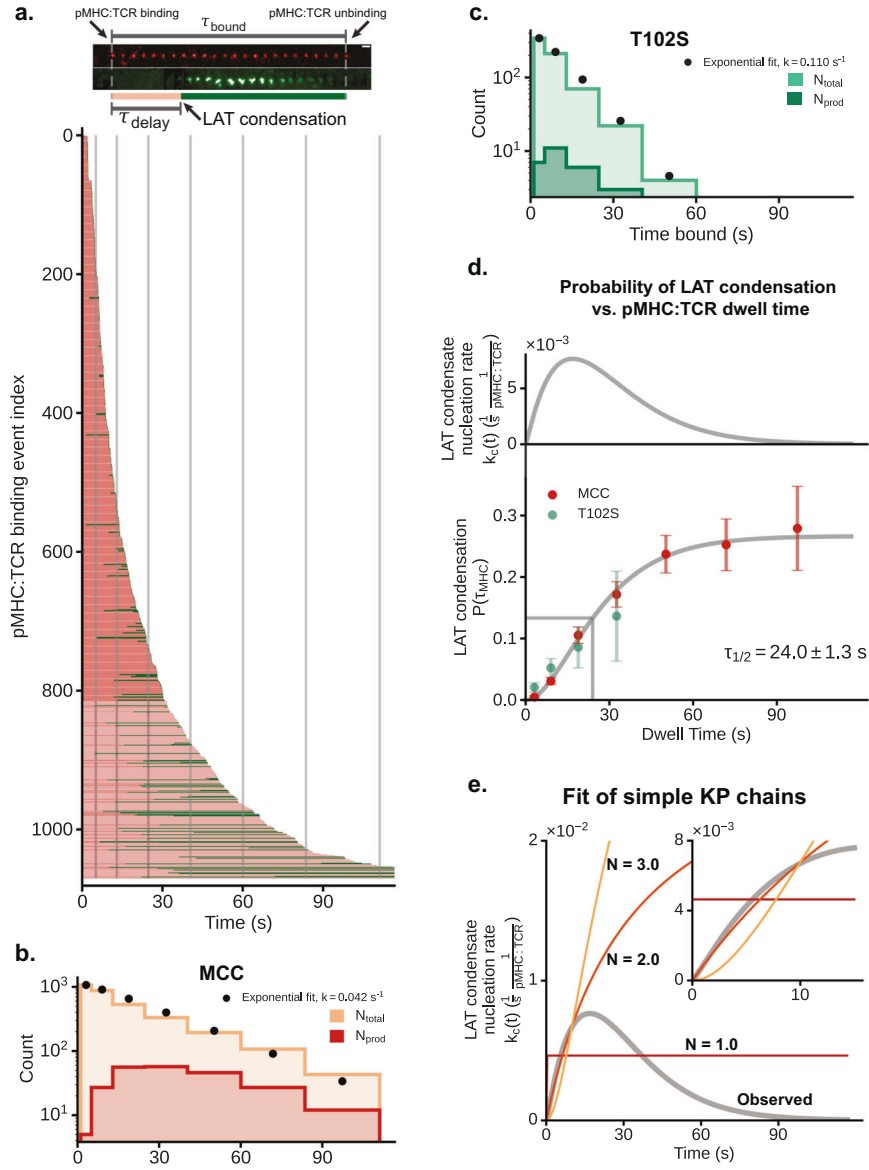

**Fig. 3 | pMHC:TCR binding dwell time modulates probability of LAT condensate formation. a** Top: time series images illustrating the temporal partitioning of the total pMHC:TCR dwell time into a delay time (pink) and a remaining productive dwell time (green). Bottom: A bar plot of a collection of 1071 pMHC:TCR binding events from 12 cells across 3 mice. Each bar represents a fully tracked pMHC:TCR binding event. The moment of condensation (if any) is indicated by the bar transitioning from pink (unproductive dwell time) to green (productive dwell time). The gray lines demarcate the bins used in the histograms for panels **b** and **c**. For **b** and **c**: Linearly increasing bin widths were used to improve the sampling rate of rare long-binding events. **b** MCC pMHC dwell time segments were aggregated according to the indicated bin widths. The number of productive dwell time segments (red) for a particular time bin was the number of binding events within that dwell time window which had at some prior time produced a LAT condensate. The total population of dwell time segments (orange) was fit with an exponential decay rate of $k_{obs} = k_{off} + k_{bleach} = \frac{1}{23.8 \text{ s}}$. The fit distribution was integrated for each bin and plotted as a black circle. **c** T102S pMHC dwell time segments were aggregated according to the indicated bin widths. The number of productive dwell time

segments (green) for a particular time bin was the of number binding events within that dwell time window which had at some prior time produced a LAT condensate. The total population of dwell time segments (aquamarine) was fit with an exponential decay rate of $k_{obs} = k_{off} + k_{bleach} = \frac{1}{9.1 \text{ s}}$. The fit distribution was integrated for each bin and plotted as a black circle. **d** Bottom: The probability of a pMHC:TCR binding event producing a localized LAT condensate as a function of dwell time. For each bin of the histograms in panels **b** and **c**, the fraction of dwell time segments that were productive is plotted as a point. The error bars were computed as the standard error of the mean for a binomial variable ($n$, the total number of binding events in the bin, and $p$ the productive fraction). The interpolating gray line was a fitted regularized gamma function with a maximal amplitude parameter. Top: The effective rate of LAT production $k_c(t)$, which is derived from the gray interpolating line in the bottom panel, is plotted along the same time axis. **e** Nucleation rates for various hypothetical simple kinetic proofreading schemes ($N = 1$, 2, or 3 steps) are plotted to illustrate key differences compared with the observed effective nucleation rate (see "$k_c(t)$ as a propensity function" in Methods for more details). Source data are provided as a Source data file.

inducing a LAT condensate after just a few seconds is small (0.05 within 10 s), it is not zero. Engagement of a sufficiently large number of TCR with pMHC, as occurs under high antigen exposure, will overcome this low probability step by sheer numbers. At low antigen densities, however, long dwelling pMHC:TCR binding events to 30 s and beyond

become important—as has been observed experimentally in single-molecule T cell activation studies[12].

The second mechanistic discovery is that the probability function for an individual pMHC:TCR complex to induce a LAT condensate plateaus at a maximum of only 0.25 (Fig. 3d, lower panel). The fact that

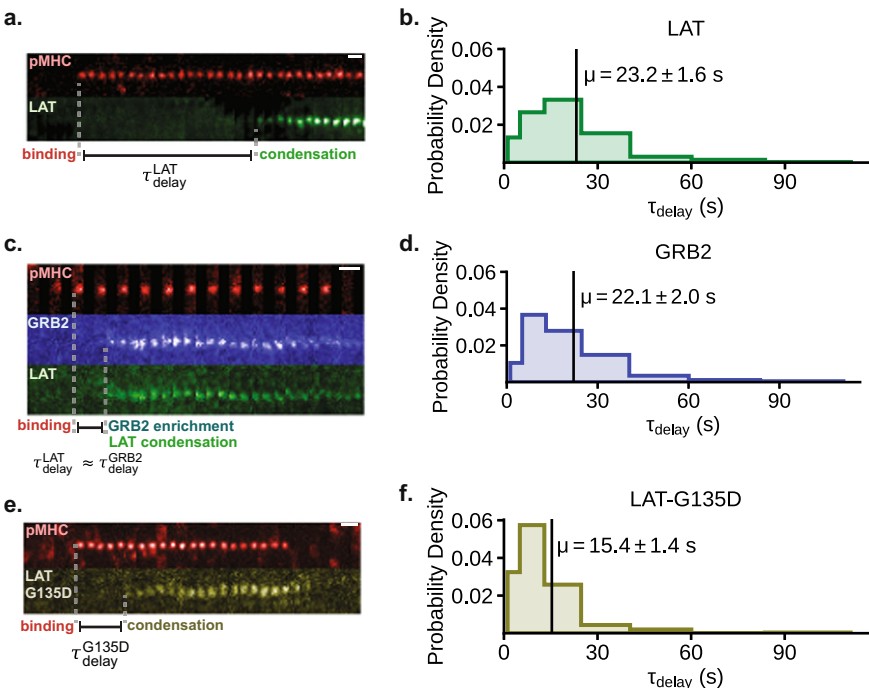

**Fig. 4 | LAT condensation occurs after an extended delay.** Primary mouse T cells expressing different constructs were deposited onto bilayers with 0.1–0.2 molecules μm⁻² of MCC(Atto647) pMHC. For the histograms, linearly increasing bin widths were used to improve the sampling rate of rare long-delay events. Scale bars are 1 μm. **a** TIRF images of LAT-eGFP condensation within T cells in response to a single pMHC:TCR binding event. LAT condensation occurs near the binding event after a long delay, $\tau_{delay}^{LAT}$, measured relative to the moment of pMHC:TCR binding. **b** Histogram of an ensemble LAT condensate delay times (115 binding events, from 18 cells across 4 mice). **c** TIRF images of LAT-mScarleti and mNeonGreen-GRB2 clustering within T cells in response to a single pMHC:TCR

binding event. Delay times of LAT clustering, $\tau_{delay}^{LAT}$, and GRB2 clustering, $\tau_{delay}^{GRB2}$, relative to binding were equivalent within the resolution of this experiment (<2 s). **d** Histogram of an ensemble GRB2 clustering delay times from T cells expressing only mNeonGreen-GRB2 (75 binding events, from 21 cells across 4 mice). **e** TIRF images of LAT(G135D)-eGFP condensation within T cells in response to a single pMHC:TCR binding event. **f** Histogram of an ensemble of delay times between pMHC:TCR binding and LAT(G135D) condensation (102 binding events, from 17 cells across 3 mice). Related data are in Supplementary Fig. 4. Source data are provided as a Source data file.

this success probability plateaus significantly below a value of one indicates the presence of some form of localized negative feedback specifically timed relative to each individual binding event. This is clearly shown by examining the momentary effective rate constant, $k_c(t)$, for formation of a LAT condensate, which is the probability per unit time of a LAT condensation event occurring a time $t$ after initial binding of pMHC to TCR. We note that $k_c(t)$ is the *propensity function* as defined in the context of stochastic kinetics[77] for the discrete process of LAT condensate nucleation. The measured success probability function, $P_{LAT}(\tau_{MHC})$, is related to $k_c(t)$ through the following integral equation: $P_{LAT}(\tau_{MHC}) = \int_0^{\tau_{MHC}} k_c(t)(1 - P_{LAT}(t))dt$, which on differentiation yields $k_c(t) = \dot{P}_{LAT}(t)/(1 - P_{LAT}(t))$. The functional form of $k_c(t)$ extracted from success probability measurements (Fig. 3d, upper panel) exhibits an initial rising phase, over the first ≈15 s, then begins declining and reaches essentially zero after 60–90 s. Practically, this indicates that a pMHC:TCR complex can only produce a LAT condensate within the first minute or so of existence. Beyond this time, even if the TCR remains engaged with pMHC, there is essentially no chance of a condensate forming. This is a single receptor level observation. The T cell itself is not shut off, and newly formed pMHC:TCR can still produce LAT condensates following the same success probability function, with the clock started for each TCR at the moment it binds pMHC. Some plausible mechanisms for how this relative timing of localized negative feedback may be achieved in T cells are detailed in Supplementary Discussion 2.

### Comparison with classic kinetic proofreading schemes
In its most basic form, kinetic proofreading for antigen discrimination by TCR has been considered as a sequence of transitions beginning

with pMHC binding to TCR. The TCR doesn't achieve activation until the final step and if pMHC unbinds before this point, activation is not achieved[8,17]. If the intermediate steps have similar rates, the strength of kinetic proofreading—referring to how sharply defined is the transition between activating and nonactivating ligands—can be roughly measured by the number of steps. Several studies of TCR and related systems have sought to infer this effective number of proofreading steps from downstream signaling parameters measured over the whole cell, such as diacylglycerol production[78], Ca²⁺ flux[79] and IL-2/CD69 expression[19].

Kinetic proofreading processes are often examined in terms of a steady state rate of activation of a downstream signaling event[8,19,80]. A more complete description of a kinetic proofreading mechanism is provided by the delay time distribution between initial ligand engagement of the receptor and the subsequent activation event, from which the propensity function ($k_c(t)$) for successful activation can be determined (see $k_c(t)$ as a "Propensity function" in Methods). This analysis enables direct comparison of the measured $k_c(t)$ for LAT condensation with predictions for various simple kinetic proofreading mechanisms. When there are no intermediate states ($N = 1$ steps), $k_c(t)$ is constant and there is no kinetic proofreading. For $N$ greater than one, $k_c(t)$ is a rising function that asymptotically approaches a maximum value when kinetic proofreading is satisfied. At early time points (<13 s), before substantial negative feedback has kicked in, the experimental data agrees well with a simple kinetic proofreading mechanism with $N = 2.0$ and an intermediate step rate of 0.03 s⁻¹, though this is not strongly distinguished from mechanisms with more steps (Fig. 3e). The marked decline in $k_c(t)$ at longer times observed in the experimental data is not predicted by classic kinetic proofreading mechanisms.

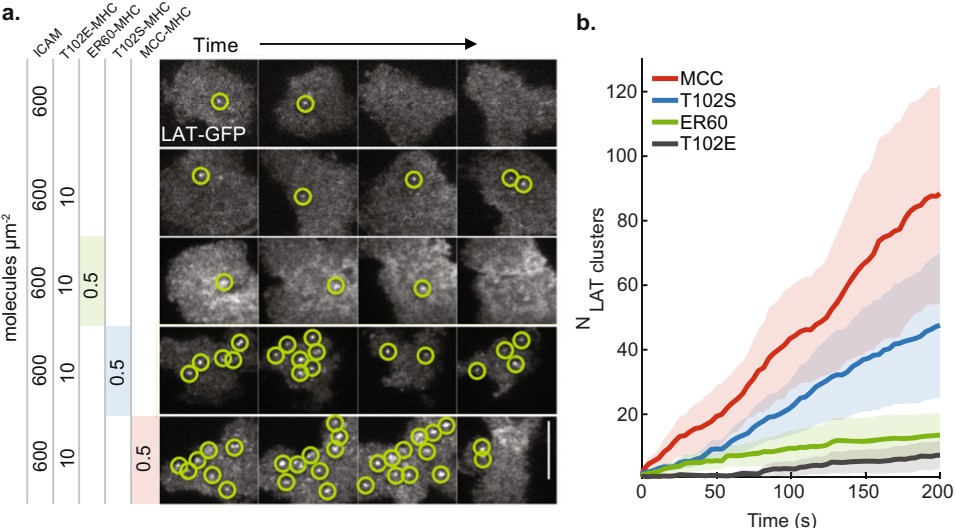

**Fig. 5 | The number of LAT condensates correlates with ligand potency. a** TIRF images over time of primary mouse T cells deposited onto supported membranes containing 600 molecules µm⁻² ICAM-1 and one of four distinct peptides at the indicated concentrations: MCC(Atto647), T102S(Atto647), ER60(Atto647), or T102E. **b** cumulative time-series for the number of distinct LAT condensates observed over a sample of cells. From most productive to least: MCC (red), T102S (blue), ER60 (green), and lastly T102E (gray). Compare to Supplementary Fig. 2g. The mean was plotted as a solid dark line, while the lighter-colored error bands were derived from ±1 standard deviation. Scale bar is 5 µm. Source data are provided as a Source data file.

## LAT condensation occurs after an extended delay

For productive binding events, LAT condensation occurs after a distinctively long delay from the originating pMHC:TCR binding event. When the LAT condensation begins, the growth rate rapidly transitions from zero to rates near 100 LAT molecules per second (see Fig. 1h and Supplementary Fig. 4a). This growth is sustained as hundreds of LAT molecules join the condensate, reaching a maximum after $9.3 \pm 6(SD)$ s (Supplementary Fig. 4b). The abrupt transition to LAT condensation establishes a well-defined delay time ($\tau_{delay}$) between the originating pMHC:TCR binding event and nucleation of LAT condensation (Fig. 4a). A histogram of 115 delay times measured from productive MCC pMHC:TCR binding events implies a broad distribution with a mean delay time of $23.2 \pm 1.6(SE)$ s (Fig. 4b). These measurements have a time resolution of 2 s, which is set by the time-lapse used in the image sequences. A collection of 91 trajectories of productive pMHC:TCR binding events can be visualized in Supplementary Fig. 4c. Control experiments confirm that the identity of the fluorophore on the LAT has no effect on LAT condensation (see LAT-mCherry condensation in Supplementary Movie 7) nor was the delay time affected by LAT expression levels (Supplementary Fig. 4f).

We next examine the cause of the delay to LAT condensation by measuring the localized concentration of phosphorylated LAT prior to condensation. Grb2 binds several phosphorylated tyrosine residues on LAT (primarily Y175, Y195, and Y235 in mouse), and does so independently of LAT condensation. Grb2 has been used as a precision probe for detailed kinetic studies of LAT phosphorylation[38,81] and here we used Grb2 to monitor LAT phosphorylation in T cells. In this experiment, primary AND T cells were transduced with a bicistronic P2A vector containing LAT-mScarleti and mNeonGreen-Grb2. TIRF imaging experiments were performed to simultaneously monitor single pMHC:TCR binding events, LAT condensation, and Grb2 recruitment in three channels. Enrichment of Grb2 was always observed simultaneously with LAT condensation with generally no detectable sustained Grb2 enrichment prior to condensation (Fig. 4c and Supplementary Fig. 4d). To rule out potential artifacts from incomplete cleavage of the P2A peptide between the two proteins, we performed further experiments with a LAT(4F)-mCherry and mNeonGreen-GRB2 linked by P2A peptide. The LAT(4F) mutant has all three primary Grb2 tyrosine binding sites (Y175, Y195, Y235) as well as the PLC-γ1 site (Y136)

mutated to phenylalanine. The mutant LAT(4F) failed to participate in any condensates while the simultaneously expressed Grb2 was observed to condense with endogenous LAT(WT) (Supplementary Fig. 4e). As an additional control experiment, monocistronic mNeonGreen-Grb2 was also used to compile a histogram of Grb2 enrichment delay times, which provide an alternative measure of LAT condensation (Fig. 4d). The resulting distribution, with a mean delay time of $22.1 \pm 2.0(SE)$ s, is nearly identical to the measured distribution using LAT imaging as the readout (Fig. 4b).

These results indicate that densities of sustained phosphorylated LAT remain undetectable above background prior to condensation. Based on the detection limit in these imaging experiments, fewer than ≈20 Grb2 molecules are sustainably localized in the vicinity of the pMHC:TCR complex prior to nucleation of LAT condensation, though momentary fluctuations certainly occur (see Supplementary Fig. 4g and "Estimation of Grb2 detection limit" in Methods). Competition from phosphatases dephosphorylating LAT and diffusion of LAT away from the active pMHC:TCR complex are likely responsible for limiting localized accumulation of phosphorylated LAT. This observation rules out one possible cause of the delay between pMHC:TCR binding and LAT condensation: that a high localized density of phosphorylated LAT must accumulate before the condensation phase transition can occur. Rather, it appears that a relatively low-density fluctuation in phosphorylated LAT from the competing kinase-phosphatase reactions themselves is the nucleating event. This is followed by a rapid accumulation of phosphorylated LAT (and associated Grb2) in the growing condensate, which is evidence for some form of positive feedback (such as phosphatase exclusion from the LAT condensate[39]). This conclusion is further supported by the broad distribution of measured delay times; some LAT condensates form within a few seconds of the originating pMHC:TCR binding event while others form after more than a minute.

## PLC-γ1 binding site on LAT controls condensation delay time

Although the delay time to LAT condensation is insensitive to both LAT and Grb2 expression levels, we find that a LAT point mutation that modulates phosphorylation kinetics by ZAP-70 substantially reduces the delay time. In human wild-type LAT, the glycine immediately upstream of Y132, the PLC-γ1 binding site, makes Y132 a poor substrate

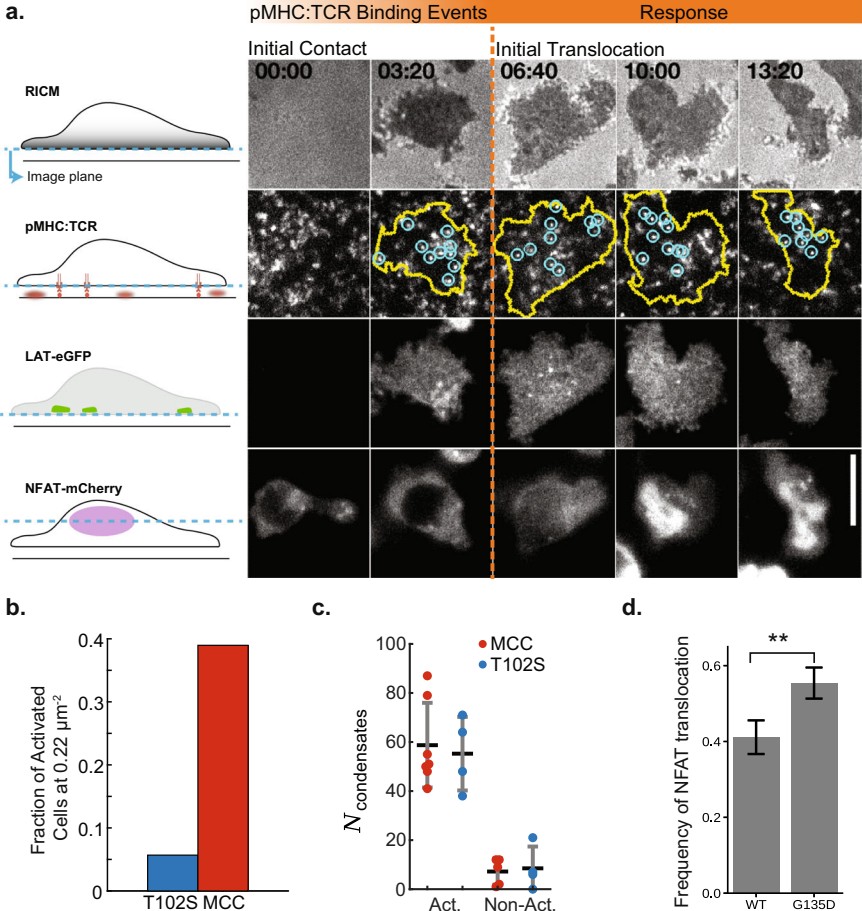

**Fig. 6 | Modulating LAT condensate delay time alters antigen discrimination thresholds. a** Images of key molecular steps leading to NFAT translocation through time within primary T cells. The cells were deposited onto supported membranes containing ICAM-1 and MCC(Atto647) pMHC. First row, RICM detects cell spreading and ensures the T cell has good contact with the supported membrane. Second row, TIRF images of individual pMHC:TCR binding events (blue) are recorded within the cell perimeter (yellow). Third row, TIRF images of LAT-eGFP is monitored for condensation events. Fourth row, epifluorescence images of NFAT to track moment of translocation. Scale bar is 5 µm. **b** Bar plot showing the frequency of T cells that undergo NFAT translocation for either T102S ($n_{cells}$ = 27) or MCC ($n_{cells}$ = 24) at a fixed peptide-MHC density of

0.22 molecules µm$^{-2}$. **c** Plot showing the number of distinct LAT condensates formed prior to NFAT translocation when exposed to different peptides. For $n$ = 12 cells exposed to MCC functionalized bilayers, 7 activated with $\mu$ = 59 ± 19 SEM distinct LAT condensates, while 5 failed to activate after 300 s with $\mu$ = 8 ± 5 SEM. Similarly, for $n$ = 8 cells exposed to T102S functionalized bilayers, 4 activated cells had $\mu$ = 56 ± 18 SEM distinct LAT condensates, while 4 that failed to activate after 300 s had $\mu$ = 10 ± 9 SEM. **d** Bar plot showing the frequency of NFAT translocation for T cells on T102S-MHC bilayers (at 1.1 molecules µm$^{-2}$) after 30 min for WT LAT ($n_{cells}$ = 124, $\mu$ = 0.41 ± 0.04 SEM) or LAT-G135D ($n_{cells}$ = 148, $\mu$ = 0.55 ± 0.04 SEM). The $p$-value was 0.02 as determined by a two-sided t-test. Source data are provided as a Source data file.

for ZAP-70 compared to Y171 and Y191. By substituting the glycine for a negatively charged aspartate, the rate of Y132 phosphorylation by ZAP-70 dramatically increases[28]. We performed the homologous mutation in mouse LAT, G135D, in order to enhance the phosphorylation rate of Y136. Primary T cells expressing LAT(G135D)-eGFP over a background of endogenous wild-type LAT continued to exhibit LAT condensation events (observed by their incorporation of the labeled LAT(G135D) mutant) in response to single pMHC:TCR binding events (Fig. 4e). However, the delay time between pMHC:TCR binding and LAT(G135D) condensation was dramatically reduced (Fig. 4f). The resulting distribution, with a mean delay time of 15.4 ± 1.4(SE) s, was narrower than that of LAT(WT); and the acceleration was achieved despite the presence of large amounts of endogenous wild-type LAT.

**The relation between LAT condensates and T cell activation**
LAT condensation (referred to as clustering in older publications) has long been recognized as a critical element of T cell activation by TCR[35,36,82]. However, most prior work has examined bulk LAT condensation in response to high antigen exposure, where hundreds to thousands of times more TCR are activated than are necessary to

trigger a full T cell response[39,67,70,83]. Here we examine the low antigen end of the spectrum to measure how the sparsely distributed LAT condensates stemming from individual pMHC:TCR binding events relate to T cell activation. We first examine LAT condensation in response to a panel of altered peptides (MCC, T102S, ER60, and T102E), spanning a range of mean dwell times and corresponding potencies[14,84]. As seen in Fig. 5, the rate and number of LAT condensate nucleation events are generally consistent with known potencies for these pMHC[83] (see "Ligand potency for LAT condensation" in Methods). In the cases of both MCC and T102S, more LAT condensation events were observed over the entire cell than simple extrapolation from single TCR measurements would predict. Some of these extra LAT condensates represent inclusion of other types of LAT clusters not directly produced from pMHC:TCR binding events (e.g., VAMP-7 containing LAT vesicles as noted above) in these measurements. However, even when other LAT clusters and potential rebinding of pMHC are considered, T102S pMHC stimulation still produces disproportionately more LAT condensates than MCC pMHC (see "Extrapolation from $P(\tau_{pMHC})$" in Methods and Supplementary Figs. 5a, b). Close examination of the T102S pMHC:TCR binding events and

associated LAT condensates implies a type of correlation that may allow the shorter-dwelling T102S pMHC ligand to potentiate LAT condensation events (see Supplementary Fig. 5c and Supplementary Discussion 3).

To quantify the relation between LAT condensation and T cell activation, we simultaneously monitored the formation of LAT condensates and NFAT translocation (as a readout of $Ca^{2+}$ activation[85–87]) in response to pMHC:TCR binding events (Fig. 6a). For these experiments, T cells were incubated on bilayers with ICAM-1 and either MCC pMHC or T102S pMHC at 0.22 molecules $\mu m^{-2}$ and were monitored for 12 min, after which we measured the final activation state (Fig. 6b). Even though a smaller proportion of cells activated in response to T102S pMHC, those cells which did activate experienced a similar number of LAT condensates as cells activating in response to MCC pMHC (Fig. 6c). This data indicates that it takes 45–75 LAT condensation events within 12 min to generate sufficient signal to induce $Ca^{2+}$ flux and subsequent NFAT translocation. The actual minimum number of LAT condensates may be lower due to the lag time of NFAT translocation. Finally, we assess how the delay time between LAT condensation and pMHC:TCR binding affects antigen discrimination using the LAT(G135D) mutant. The G135D mutation has been shown to enhance both $Ca^{2+}$ and ERK activity in response to weaker agonists, particularly at low stimulation thresholds[51]. Results comparing NFAT translocation rates between T cells expressing LAT(WT) or LAT(G135D) show that the presence of LAT(G135D) increased NFAT translocation rates by ≈15% (Fig. 6d; Supplementary Movie 8). These experiments were done using the weaker T102S pMHC agonist at its activation threshold density, 1.1 molecules $\mu m^{-2}$ [12]. While the observed effect is modest, it provides important evidence that alterations in LAT condensation kinetics, which occur on the 20–30 s timescale, still change the T cell responsiveness to significantly shorter dwelling pMHC ligands.

## Discussion

Certain output parameters of the T cell activation response, including NFAT and Erk activation as well as cytokine production, exhibit essentially binary behavior at the single cell level[10,87,88]. When observing these parameters, exposure to higher antigen pMHC densities may increase the probability of an individual cell activating—or, equivalently, an apparently stronger activation response from a population of T cells—but it does not change signal strength at the single-cell level. By contrast, many molecular intermediates in the TCR signaling pathway, including TCR phosphorylation, ZAP-70 recruitment, LAT phosphorylation, and LAT condensation exhibit a much more linear dose-response behavior with respect to the copy number of agonist pMHC molecules at moderate to high levels[44,72,89]. Titrating down to single molecule levels, TCR and ZAP-70 activation remain essentially stoichiometric with agonist pMHC[13] whereas LAT exhibits substantially different behavior. The stochastic formation of discrete LAT condensates (containing hundreds of LAT molecules) in response to individual pMHC:TCR binding events that we report here represents a distinct transition in the propagation of signal downstream from TCR. This implicates not only a major amplification step, but also a thresholding mechanism that quenches downstream signaling from shorter-lived, but still potentially ZAP-70 kinase active, pMHC:TCR complexes. The further observation that a LAT condensate's size and lifetime are not correlated with the binding dwell time of its originating pMHC:TCR complex is indicative of a discretization mechanism, by which a continuum of pMHC:TCR binding dwell times is parsed into a binary success or failure output. This signal digitization would apply to signals dependent on formation of the LAT signaling complex; it is also possible some TCR signaling activity is independent of this—such as tonic signaling activity from very short-lived self pMHC:TCR interactions[90–92].

Clustering and assembly of molecules in the TCR signaling pathway has been a topic of intense interest, from both an immunological[44,70,72] and biophysical perspective[15,55,93–95]. Foundational experiments on LAT have mapped the presence of multiple distinct Grb2, Gads, and PLC-γ1 phosphotyrosine binding sites as required elements for downstream propagation of signaling from activated TCR[35,96,97]. Key realizations from this early work are that multiple pY sites needed to be on the same LAT molecule (e.g., multivalency is required) and that all these molecules condense into a LAT-scaffolded signaling microcluster (here referred to as a LAT condensate) in T cells[29,82]. These early studies were almost invariably done with high agonist pMHC stimulation or antibody stimulation (which involves artificial TCR crosslinking). Here we have presented experimental results that zoom in on individual pMHC:TCR binding events for a live, superresolution view of the initiation of LAT condensation in T cells.

The single-molecule imaging experiments show signature behavior of a protein condensation phase transition that is not detectable in bulk observations. We observe that discrete LAT condensates form in response to individual pMHC:TCR binding events and that LAT condensation occurs abruptly, and after a well-defined delay time from the originating pMHC:TCR binding event. The delay times are broadly distributed, with some LAT condensates forming within a few seconds of the pMHC:TCR binding event while others may delay up to a minute. In all cases, there is no detectable build-up of localized pLAT prior to condensation. Collectively, these observations are consistent with a phase transition nucleation process controlled by fluctuations in phosphorylation reaction kinetics; the competing kinase-phosphatase reactions controlling LAT phosphorylation provide the nucleating event[98]. These features of the phase transition are readily visible at low agonist pMHC densities because individual binding events remain well separated and can be unambiguously tracked for extended times. At higher pMHC densities, multiple LAT condensates originating from different pMHC:TCR binding events appear superimposed and unsynchronized in the composite LAT condensation response (Fig. 1b, c and Supplementary Fig. 1c). This creates an illusion of a more continuous growth process while the underlying mechanism is likely to still involve the discrete phase transition we observe at the single-molecule level.

A critical remaining question is what are the qualitative differences between a kinetic nucleation phase transition process and a more linear growth process with respect to signal propagation through LAT? Two key observations provide a clue: (i) a single activated pMHC:TCR complex is not able to build up a localized concentration of pLAT prior to condensation; and (ii) LAT condenses abruptly once nucleation is achieved. This combination of features implicates an initially strong resistance to signal propagation (e.g., LAT phosphorylation) followed by positive feedback that favors LAT phosphorylation once condensation has been initiated. This type of initial resistance followed by positive feedback is an effective noise filtering mechanism (commonly found in electronic devices such as avalanche photodiodes used for single photon detection[99]). If LAT condensation were to follow a more linear phosphorylation and growth process at the single molecule level, this type of noise-filtering mechanism would be defeated. A second insight is provided by recent studies on Ras activation by SOS in LAT condensates that identified a strong enhancement of SOS activity in the condensed state[40]. Mechanistically, multivalency in the condensate retains SOS at the membrane for extended periods of time, providing sufficient time for autoinhibition release. It is plausible that a similar mechanism may govern PLC-γ1 activation.

Finally, the discrete and delayed condensation of LAT tied to individual pMHC:TCR binding events provides an additional layer of kinetic proofreading for antigen discrimination. This is revealed in our measurement of the single TCR antigen discrimination function (Fig. 3d). The long mean delay time between LAT condensation and the originating pMHC:TCR binding event renders short dwelling pMHC:TCR complexes highly unlikely to produce a LAT condensate

and any downstream signals that rely on this. This effect is most pronounced at low pMHC levels, where cooperativity between multiple pMHC:TCR complexes is essentially impossible since they are so widely spaced. At higher antigen densities, we have observed instances of a spatially localized sequence of short pMHC:TCR binding events evidently contributing to a single LAT condensate (Supplementary Fig. 5c). This shows a cooperative mechanism that enables the T cell to respond to shorter dwelling pMHC ligands that are present at sufficiently high density. Effectively, the final stage of kinetic proofreading provided by the LAT condensation event is bypassed at higher antigen density since multiple pMHC:TCR may work together to achieve nucleation. Earlier stages of kinetic proofreading (e.g., Lck phosphorylation of ITAMS, ZAP-70 recruitment, and ZAP-70 activation) are still necessary and pMHC ligands that fail to pass these stages won't activate T cells at any density[2,3,18,19].

Protein condensation phase transitions are emerging as important mechanistic features in a wide range of biological processes[39,98,100–102]. Here, using single-molecule imaging methods, we have resolved signature characteristics of the LAT protein condensation phase transition in T cell activation. The results provide quantitative mapping of early signal filtering and amplification mechanisms in TCR signaling. More broadly, they imply a number of functional capabilities that are provided by the phase transition process that may also be found in other membrane signaling systems.

## Methods

### T cell culture and transduction

Primary human T cells were sourced from StemCell Technologies and were from de-identified donors with approved consent forms and protocols (Human Peripheral Blood Leukopak). The mice used in this study were the cross of (B10.Cg-Tg(TcrAND)53Hed/J) x(B10.BR-H2k2 H2-T18a/SgSnJ) from Jackson Laboratory. All animal work was performed with prior approval by the IACUC committee, Lawrence Berkeley National Laboratory Animal Welfare and Research Committee, under the approved protocol 17703. Lymph nodes and spleens of 6- to 20-week-old mice were stimulated with 1 μM MCC peptide in vitro. Both male and female mice were used within this study and were monitored for clean health prior to organ harvest. Splenocytes from the TCR(AND) mice, hemizygous for H2$^k$, were pulsed with 1 μM moth cytochrome c (MCC) peptide and cultured with the T cells for two days. The T cell blasts were treated with IL-2 from the day after harvest to the fifth day after harvest, at which point the cells were used in experiments. Activated T cells were retrovirally transduced using Platinum-Eco cell-derived supernatants. Platinum-Eco cells (Cell Biolabs, San Diego, CA) were transfected with the desired plasmid using linear, polycationic polyethylenimine (Sigma-Aldrich). After 48 h of retrovirus production, the supernatant of the PLAT-E cells was used to spinfect T cells in the presence of polybrene (4 mg/ml) on day 3 after activation with MCC peptide. Typical transduction efficiencies ranged from 10-30%. The entire population was used for imaging experiments; 0.5 million to 2 million cells were deposited onto each bilayer. Positive fluorescent cells were easily distinguished from negative cells. Overall phenotypic behavior of crawling and spreading were comparable between the two populations using RICM to ensure transduction of the various constructs did not alter basic T cell behavior.

### Protein purification

Histidine-tagged MHC class II I-E$^k$ and ICAM-1 were expressed and purified as previously described[57]. MHC II with C-terminal hexahistidine tags on both α and β chains were expressed using a baculovirus expression system in S2 cells and purified using a Ni−nitrilotriacetic acid (NTA) agarose column (Qiagen). The histidine-tagged MHC bacmid was a gift of L. Teyton (Scripps Research Institute)

and M. Davis (Stanford University). The bacmid for ICAM-1 with a C-terminal decahistidine was synthesized, and it was similarly expressed and purified in High Five cells (Invitrogen).

### DNA constructs

The murine stem cell virus (MSCV) backbone was used (similar to AddGene #91975) for retroviral transduction of primary mouse T cells. Plasmids were constructed using Gibson Assembly and cloned using XL1-BLUE (Agilent Technologies) *E. coli*.

MSCV-NFAT-mCherry was constructed from a plasmid containing a GFP fusion to the regulatory domain of the mouse NFAT1 (NFATc2) protein in a murine stem cell virus vector[85] and was a gift of F. Marangoni (Harvard Medical School). This truncated form of NFAT1 [pMSCV-NFAT1(1−460)-GFP] contains the regulatory domain that controls the nucleocytoplasmic shuttling of NFAT but lacks the DNA binding domain. A second version of the plasmid was generated to replace the GFP coding sequence with mCherry.

MSCV-mNeonGreen-GRB2 was constructed from a plasmid containing human GRB2 that was a gift from Neal Shah[51]. An mNeonGreen license and plasmid stock was purchased from Allele Biotechnology.

MSCV-LAT-eGFP was constructed from a plasmid containing cDNA for *Mus Musculus* Linker for Activation of T cells (NM_010689.3).

MSCV-LAT-eGFP-P2A-mNeonGreen-GRB2 was subcloned from the constructs above with the addition of the P2A sequence that was ordered as an oligonucleotide from Elim Biopharmaceuticals, Hayward, CA.

### Peptide synthesis and labeling

Moth cytochrome C 88–103 peptide (MCC88−103; abbreviated as MCC) and previously characterized variants[74] were synthesized and lyophilized on campus (D. King, Howard Hughes Medical Institute Mass Spectrometry Laboratory at University of California, Berkeley) or commercially (Elim Biopharmaceuticals, Hayward, CA). A short flexible linker of three amino acids and terminal cysteine was added to the C terminus for fluorophore labeling. The sequences are as follows: MCC (ANERADLIAYLKQATK), MCC(C) (ANERADLIAYLKQATKGGSC), T102S (ANERADLIAYLKQASK), T102S(C) (ANERADLIAYLKQASKGGSC), and T102E (ANERAELIAYLTQAAEK). For dye conjugation, the cysteine-containing peptide sequences were reacted with the maleimide-containing organic fluorophore of interest (Atto647N, Atto565, or Atto488; Atto-Tec GmbH, Siegen, Germany) in phosphate-buffered saline (PBS) with a trace amount of 1-propanol. The labeled peptides were purified using a H$_2$O/acetonitrile gradient on a C18 reverse-phase column (Grace Vydac, Deerfield, IL) in the AKTA Explorer 100 FPLC system (Amersham Pharmacia Biotech, Piscataway, NJ). Mass spectrometry was used to confirm the peptide identity after purification.

### Bilayer preparation

**Ni$^{2+}$ chelated bilayers.** Glass-supported lipid bilayer membranes were prepared in imaging chambers and functionalized with proteins in a manner similar to prior experiments[13,103]. At 18 to 24 h before imaging, MCC and variant peptides were loaded into MHC II I-E$^k$ at 37 °C in peptide-loading buffer [PLB−1% (w/v) bovine serum albumin (BSA) in PBS (pH 4.5) with citric acid]. Just before exposure to bilayers, dye-peptide-MHC complexes in PLB were purified using a 10,000-molecular weight cutoff spin concentrator (Vivaspin 500, GE Healthcare, Pittsburgh, PA) and 1x TBS wash [TBS; 19.98 mM tris and 136 mM NaCl (pH 7.4); Mediatech Inc., Herndon, VA], and resuspended to 300 μL. Small unilamellar vesicles (SUVs) are formed using tip sonication with the composition of 98 mole percent (mol%) 1,2-dioleoyl-snglycero-3-phosphocholine (DOPC) and 2 mol% 1,2 dioleoyl-sn-glycero-3-[(N(5-amino-1-carboxypentyl) iminodiacetic acid) succinyl] (nickel salt) (Ni2+-NTA-DOGS) (Avanti Polar Lipids, Alabaster, AL) in Milli-Q water (EMD Millipore, Billerica, MA)[103]. In addition, #1.5 25-mm-diameter round coverslips (Thomas Scientific #1217N82) were ultrasonicated in

1:1 isoporopyl/$H_2O$ and etched for 5 min in piranha solution (3:1 $H_2SO_4$/$H_2O_2$). 35-mm AttoFluor chambers (Fischer Scientific #A7816) were used to house the etched coverslips. A 1:1 mixture of SUVs and 1x TBS was introduced into the chambers, and bilayers were allowed to form through vesicle rupture for at least 30 min. After rinsing, the bilayer was activated with 100 mM $NiCl_2$ for 5 min. Samples were exchanged to a live T cell imaging buffer [LCB−1 mM $CaCl_2$, 2 mM $MgCl_2$, 20 mM Hepes, 137 mM NaCl, 5 mM KCl, 0.7 mM $Na_2HPO_4$, 6 mM D-glucose, and 0.1% (w/v) BSA]. Immediately before imaging, the bilayer was incubated for 25 min with His-pMHC (≈10 pM) and His-ICAM-1 (≈10 nM) in LCB. The bilayer was rinsed with imaging buffer after the incubation, and the His-tagged/Ni-NTA− bound proteins were allowed to equilibrate on the bilayer for 20 min. The resulting supported membranes typically display ICAM-1 at 100 to 200 $\mu m^{-2}$ and pMHC at 0.1 to 2 $\mu m^{-2}$. The chamber was equilibrated to 37 °C, and T cells resuspended in T cell imaging buffer were introduced. All imaging was done in 37 °C.

**OKT3 streptavidin bilayers.** SUVs were prepared, ruptured onto coverslips, and activated with $NiCl_2$, as above, but with a lipid composition of 98 mol% DOPC, 2 mol% Ni-NTA-DOGS, and 0.02 mol% of 1,2-dioleoyl-sn-glycero-3-phosphoethanolamine-N-(cap biotinyl) (biotin-CAP-PE) (Avanti Polar Lipids, Alabaster, AL). Streptavidin (Sigma-Aldrich, St. Louis, MO) was filtered through a 0.1 μm centrifugal filter (EMD Millipore, Billerica, MA) to remove aggregates, diluted with 1x LCB to 0.5 μg/mL, and then incubated with the bilayers for 45 min, rinsed with 1x LBC, and then incubated with biotinylated anti-human CD3 antibody (OKT3) (BioLegend # 317319) at 1 μg/mL for 35 min, followed by a rinse with 1x LCB. The bilayers were then incubated with histidine-tagged human ICAM-1 (≈10 nM) in LCB for 20 min, then rinsed with LCB. T cells were introduced as described above.

## Imaging
All imaging experiments were performed on a motorized inverted microscope (Nikon Eclipse Ti-E; Technical Instruments, Burlingame, CA) with a motorized Epi/TIRF illuminator, a motorized Intensilight mercury lamp (Nikon C-HGFIE), and a motorized stage (MS-2000; Applied Scientific Instrumentation, Eugene, OR). A laser launch with 488-, 560-, and 640-nm diode lasers (Coherent OBIS, Santa Clara, CA) was aligned into a custom-built fiber launch (Solamere Technology Group Inc., Salt Lake City, UT). For TIRF imaging, laser illumination was reflected through the appropriate dichroic beam splitter (ZT488/647rpc, Z561rdc with ET575LP) to the objective lens [Nikon (1.47, numerical aperture; 100×), TIRF; Technical Instruments, Burlingame, CA]. RICM and epifluorescent excitation were filtered through a 50/50 beam splitter or band-pass filters (D546/10×, ET470/40×, ET545/30×, and ET620/60×). All emissions were collected through the appropriate emission filters (ET525/50M, ET600/50M, and ET700/75M) and captured on an EM-CCD (iXon 897DU; Andor Inc., South Windsor, CT). All filters were from Chroma Technology Corp. (Bellows Falls, VT). All microscope hardware was controlled using MicroManager[104].

Laser power was measured at the sample plane with a Newport Power Meter Model 1918-R controller and a 918D-SL-0D2R.

Single-molecule TIRF images are taken at the same power dosage (*power*\**exposure*), either by high laser illumination intensity (8 mW) and short exposure time (25 ms) or by low laser illumination intensity (0.4 mW) and long exposure time (500 ms). Single-molecule TIRF images of long exposure time (500 ms) were collected every 2 to 4 s to localize single pMHC-TCR using a laser illumination intensity of 0.4 mW. LAT-eGFP was monitored at 0.4 mW every 2 to 4 s. When cells were imaged to measure LAT condensation only, the cells were imaged as they landed, or within 60 s of landing, for a duration of 300 s. Cells imaged for NFAT-mCherry was done by epifluorescence using 150-ms exposure time every 50 s at 3 and 6 μm above the coverslip to monitor NFAT dynamics. NFAT monitoring continued for 30 min unless otherwise stated.

Bleach rates for MCC(Atto647) pMHC were measured by incubating pMHC using 1x PBS in an imaging chamber containing a piranha etched glass coverslips (no bilayer) using pMHC volumes that would typically generate 0.1 $\mu m^{-2}$ pMHC. This creates spatially fixed pMHC. The imaging conditions were set identical to those used for cellular acquisitions. The number of spots within the centrally illuminated region (approximately 350 × 350 pixels) were tracked through time. Only tracks that appeared in the first frame and were fully immobile were counted. The population of spots through time was fit with an exponential decay. Only the first 150 frames are used for the fit as there tends to be changing decay dynamics at different time scales (likely due to uneven illumination), and typically cells are only imaged for 150–200 frames.

There is a tradeoff between temporal resolution and bleach rate. For the 2 s time-lapse acquisitions that better capture the moment of LAT condensation, a commonly observed bleach rate was $k_{bleach} \approx \frac{1}{99sec}$, which when compared to the dwell time of MCC, $k_{off} \approx \frac{1}{45s}$, leads to a $\frac{k_{bleach}}{k_{bleach} + k_{off}} = \%31$ chance that the failure of a pMHC to appear in the next frame was due to bleaching, as opposed to unbinding.

## Western blot
The general procedure was adapted from Lo et al.[89]. Primary mouse CD4+ T cells were transduced with MSCV bearing LAT-eGFP. After 48 h, cells were lysed by direct addition of 10% NP-40 lysis buffer to a final concentration of 1% NP-40 (containing the inhibitors 2 mM NaVO4, 10 mM NaF, 5 mM EDTA, 2 mM PMSF, 10 μg/ml aprotinin, 1 μg/ml pepstatin, and 1 μg/ml leupeptin). The lysates were placed on ice and centrifuged at 13,000 × $g$ to pellet cell debris. The supernatants were run on NuPAGE 4–12% Bis-Tris protein gels (Thermo Fisher) and transferred to nitrocellulose using a NuPAGE XCell II Blot Module. Membranes were blocked with TBS-T buffer containing 3% BSA, then probed with primary LAT antibody (Cell Signaling Technology 9166P) overnight at 4 °C. The following day blots were rinsed and incubated with secondary IRDye 680RD Donkey anti-Rabbit IgG (Li-cor 926-68073). And visualized using the Odyssey Li-cor Classic and with the Odyssey One-Color Protein Molecular Weight Marker.

## Simulation of whole cell integrated LAT signal
A simple model is constructed to illustrate the effect of sample size on observed delay time, it is not intended to emulate bona fide LAT responses. For a static field of potential ligand binders and an already adhered T cell, the "time until a ligand binds" can be modeled to follow an exponential distribution. In our supported membrane experiments we observe a pseudo-kinetic on-rate of $k_{on}^{cell} = 3.8 \pm 2.6 \ \mu m^2 s^{-1}$ (see "Estimated pMHC on-rate" below). Using this on-rate and a specified ligand density, we can sample when a ligand binds, $t_{on}$, from an exponential distribution with parameter $k = k_{on}^{cell} \sigma_{pMHC}$. A priori, we expect 0.022 of MCC pMHC binding events with 5cc7 TCR−which has a 6 s dwell time[13];−to produce a LAT condensate (see "A priori success probability of pMHC binding" below). Of those productive binding events, two random variables were generated, first a random delay time (from the distribution in Fig. 4b) and a "random" pulse of LAT signal. A pulse was modeled as a scaled gamma distribution function that reflected the shapes of LAT pulses seen in our experiments (compare Supplementary Fig. 1d and Supplementary Fig. 3a).

Sampling from 10 s to 1000 s of productive binding events, the superposition of the corresponding number of LAT pulses shows that as the number of sample LAT pulses increases, the moment of first detectable signal also becomes earlier (Supplementary Figs. 1d, e).

## Tracking pMHC and LAT
A combination of the standard TrackMate plugin[105] for ImageJ and a custom in-house TrackMate plugin was used. The custom plugin was specifically developed to account for the centripetal motion observed

by pMHC:TCR binding events. Despite the improved accuracy, each binding event still required manual inspection, using TrackMate's built-in track inspection tools, to eliminate occasional tracking/detection artifacts. Single-molecule trajectories were verified using single-step unbinding determined by a Bayesian change point detection algorithm[106].

A binding event trajectory and dwell time was considered valid if it did not have any branching (i.e., no splits or merges). A LAT delay time was considered valid if its associated pMHC binding event did not have any branching (i.e., no splits or merges) prior to condensation.

LAT condensates were initially tracked with TrackMate's automated workflows and then each track was manually adjusted for proper linking/detection. A pMHC binding event was considered productive if there existed a LAT condensate trajectory whose first frame of detection could be found within 250 nm of the binding event.

## Quantification and statistical analysis
Data expressed as $x \pm y$ (SD) represents a mean of $x$ and standard deviation of $y$. While (SE or SEM) refers to $y$ as the standard error of the mean. Data expressed simply as $x \pm y$, lacking SD or SE, arise from fit parameters, the error of fit was calculated from the square root of the covariance matrix when using the `curve_fit` method in the `scipy` python library. The `curve_fit` method from the `scipy` library returns a covariance matrix `pcov`, that is based on scaling the error (sigma) by a constant factor. This constant is set by demanding that the reduced `chisq` for the optimal parameters, `popt`, when using the scaled sigma equals unity. In other words, `sigma` is scaled to match the sample variance of the residuals after the fit. See https://docs.scipy.org/doc/scipy/reference/generated/scipy.optimize.curve_fit.html for more information.

The meaning of the sample size, $n$, is specified in relevant figure legends. Quantification and statistical analysis were done using Python and the following libraries – scikit-image, scipy, pandas, and numpy.

## Dwell times
Reported pMHC:TCR dwell times, $\langle \tau_{dwell} \rangle = \frac{1}{k_{off}}$, were corrected for photobleaching by measuring the bleach rate of fluorescent pMHC on supported membranes, $k_{bleach}$, and then fitting the observed dwell time distribution, $f_{obs}(t)$, to the following equation:

$$f_{obs}(t) = \left(k_{bleach} + k_{off}\right)e^{-\left(k_{bleach} + k_{off}\right)t} \quad (1)$$

## FRAP of LAT condensates
Many studies of protein condensation phase transitions use FRAP as a technique to estimate the fluidity of the system[39,107]. A feature of LAT condensates in living cells that complicates this measurement is that the condensates have definite growth and decay phases (Supplementary Fig. 2a) as well as centripetal motion towards the center of the cell. However, by intermittently imaging the cell at a fast frame rate (20 ms) and higher laser power followed by multiple seconds to allow recovery, we were able to see significant recovery in the LAT condensates regardless of whether the condensate was actively growing or decaying. Assuming the recovery was entirely reaction-limited, the estimated off-rate for LAT within the condensate was $k_{off} = 0.24\,s^{-1}$, though this is likely a lower-bound estimate.

## Ligand potency for LAT condensation
Data presented in Fig. 5 originated from T cells that were exposed to supported membranes containing 600 molecules $\mu m^{-2}$ of ICAM-1 and 0.5 molecules $\mu m^{-2}$ pMHC after loading MHC with one of the peptides from the panel. In most conditions, high amounts of the null T102E pMHC (10 molecules $\mu m^{-2}$) were included as background for the tested pMHC to mimic the abundant self pMHC that exists on antigen-presenting cells in vivo. LAT condensation events were tracked over

time by TIRF imaging using the LAT-eGFP construct. The null T102E pMHC + ICAM-1 condition itself did not produce any LAT signal over background levels observed with ICAM-1 only conditions.

## NFAT translocation
NFAT translocation to the nucleus occurs downstream of $Ca^{2+}$ activation in T cells and has been successfully used as a read-out of early T cell activation[85–87]. Here, we expanded on an assay previously developed by our lab that mapped patterns of pMHC:TCR binding to NFAT translocation[12] by incorporating LAT-eGFP into the expression vector using a P2A peptide.

NFAT translocation was decided by measuring the fluorescence intensity in the cytosol and nucleus. All images with the masked regions were inspected. The time point at which the mean nuclear NFAT intensity is first detected to be above the mean cytosolic intensity was set as the time point of initial NFAT translocation.

## Extrapolation from $P_{LAT}(\tau_{pMHC})$
From Fig. 5 we note that after 200 s, the longer dwelling MCC peptide ($\tau_{dwell} \approx 45\,s$) produced $83 \pm 35$ (SD) distinct LAT condensates, while the shorter dwelling agonist T102S ($\tau_{dwell} \approx 10\,s$) and non-agonist ER60 ($\tau_{dwell} \approx 1\,s$) produced $42 \pm 20$ (SD) and $8 \pm 2$ (SD) LAT condensates, respectively. These numbers will be compared to a rudimentary extrapolation from the single TCR activation function. This extrapolation begins with estimating the probability that a binding event produces a LAT condensate.

**A priori success probability of pMHC binding.** The success probability of a ligated TCR (Fig. 3d), as a function of dwell time, can be used to approximate the LAT response to peptides of differing dwell times at low peptide density. This is done by multiplying the dwell time distribution of the pMHC by the success probability function (Fig. 3d) and integrating over time.

$$p\left(k_{off}\right) = \int_{0}^{\infty} k_{off}e^{-k_{off}t}P_{LAT}(t)dt \quad (2)$$

For MCC, which has an exponentially distributed dwell time of ≈4 s, the a priori success probability of a binding event producing a LAT condensate is 0.135. Similarly, the a priori success rates for T102S and ER60 binding are 0.044 and 0.003, respectively. Given the same number of spatially isolated binding events, MCC would be expected to produce $\frac{0.135}{0.044} = 3.07$ times the number of LAT condensates as T102S. However, experimentally, for an equal density of pMHC, only a 2-fold difference in the number of LAT condensates is observed (Fig. 5). Next, we consider potential differences in the number of binding events.

**Estimated pMHC on-rate.** The on-rate of pMHC can change throughout the cell landing and activation process[14]. To estimate the on-rate with the presence of photobleaching we consider the simple kinetic scheme:

$$R + L \rightarrow C \quad (3)$$

In which receptor ($R$) binds to ligand ($L$) to form a complex ($C$). For a TCR density $\sigma_R(t)$, and a *visible* ligand density that exponentially decreases due to bleaching, $\sigma_L(t) = \sigma_L e^{-k_b t}$, where $\sigma_L$ is the initial ligand density, then the rate of changing complex density ($\sigma_C$) is modeled by the following rate equation:

$$\frac{d\sigma_C}{dt} = \frac{1}{A_{cell}(t)}\frac{dN}{dt} = k_{on}(t)\sigma_R(t)\sigma_L e^{-k_b t} \quad (4)$$

Where $N$ is the total number of visible binding events observed over time $t$. To proceed, we instead parameterize a rate equation that uses static mean approximations, and we solve for a pseudo kinetic on-rate, $k_{on}^{cell}$ that absorbs the mean cell area and TCR density terms. Integrating over time and solving for $k_{on}^{cell}$ gives:

$$k_{on}^{cell} = \frac{N}{\sigma_L \left( \frac{1}{k_b} - \frac{1}{k_b} e^{-k_b t} \right)} \tag{5}$$

For a collection of 15 T cells expressing TCR(AND), across 3 mice, the parameters $N, k_b, \sigma_L$, and $t$ were all determined and the resulting $k_{on}^{cell}$ of MCC pMHC binding to T cell was found to be $k_{on}^{cell} = 3.8 \pm 2.6 \,\mu m^2 s^{-1}$. T102S had a similar rate. As an example, for a ligand density of $0.1 \,\mu m^{-2}$, over 60 s one can expect $3.8 * 0.1 * 60 \approx 23$ binding events from MCC pMHC. Note that this rate constant was calculated over low ligand densities, $0.1 – 0.6 \,\mu m^{-2}$, and will lose applicability under saturating ligand densities.

**Empirical LAT surplus.** For Fig. 5, T cells expressing TCR(AND) were deposited onto supported lipid bilayers displaying $\approx 600 \,\mu m^{-2}$ ICAM-1 and $0.5 \,\mu m^{-2}$ of pMHC. To account for photobleaching, rebinding, and potential depletion of local pMHC, a simplified Gillespie simulation[77] was performed to estimate the number of binding events over 200 s. The parameters used were $k_{on}^{cell}$ (measured above), the off-rate of the peptide, and the diffusion coefficient of free pMHC $(0.55 \,\mu m^2 s^{-1})$. From an ensemble of 250 simulated trajectories, T102S had an average of 375 binding events over 200 s (Supplementary Fig. 5a), while MCC had an average of 355 binding events (Supplementary Fig. 5b). This suggests that diffusion is fast enough, at this density, to prevent significant depletion effects.

The ratio of empirical to predicted numbers of LAT condensates for MCC was $\frac{83}{355 * 0.135} = 1.7$; and for T102S the surplus ratio was $\frac{42}{375 * 0.044} = 2.5$.

## $P_{LAT}(\tau_{MHC})$ and LAT delay times

The combination of dwell time from MCC-MHC:TCR binding and the imaging conditions (power, exposure, etc.) gives rise to an exponential decay of "observed dwell times".

$$n_{dwell}^{obs}(\tau \geq t) = N^{obs} e^{-k_{obs} t} \tag{6}$$

Where $N^{obs}$ is the total number of observed pMHC:TCR binding events, and $k_{obs} = k_{off} + k_{bleach}$. The empirical frequency distribution for [6] is the light orange histogram present in Fig. 3b. If follows that the observed frequency of *productive* pMHC:TCR dwell times has the functional form:

$$n_{dwell}^{prod}(t) = n_{dwell}^{obs}(\tau \geq t) P_{LAT}(t) \tag{7}$$

Where $P_{LAT}(t)$ is the cumulative probability that an individual pMHC:TCR binding event bound for time $t$ has produced a localized LAT condensate. Both [6] and [7] depend on imaging conditions. The empirical frequency distribution for [7] is the red curve present in Fig. 3b. By dividing $n_{dwell}^{prod}(t)$ by $n_{dwell}^{obs}(\tau \geq t)$, we recover $P_{LAT}(t)$ (Fig. 3d). Moreover, $k_{obs}$ now merely affects the sampling rate of $P_{LAT}(t)$, not the shape of $P_{LAT}(t)$.

The histogram of delay times between pMHC:TCR binding and LAT condensation (Fig. 4b) arises from the following functional form:

$$L_{delay}^{obs}(t) = n_{dwell}^{obs}(\tau \geq t) \dot{P}_{LAT}(t) \tag{8}$$

Where $\dot{P}_{LAT}(t)$ is the probability distribution function of delay time from pMHC:TCR binding until LAT condensation is observed. The number of delay times observed occurring at time $t$, $L_{delay}^{obs}(t)$, is $\dot{P}_{LAT}(t)$ multiplied by the number of binding events that are known to exist for

*at least* a duration $t$. As such [8], is affected by $k_{obs}$, and for meaningful comparisons of delay time distributions, the imaging conditions and peptide-MHC must be the same, as they are in Fig. 4b, d, and f. What is plotted in Fig. 4b is $L_{delay}^{obs}(t)$ ratioed to the total number of LAT condensates observed to originate from pMHC:TCR binding.

## $k_c(t)$ as a propensity function

The momentary probability per unit time of a LAT condensate forming a time $t$ after initial binding of pMHC to TCR is the propensity function $(k_c(t))$ for the stochastic process of LAT condensate nucleation. The propensity function is also known as a hazard rate, $h(t)$, and is the probability density that an event occurs at time $t$, given that it has not yet occurred. For the measured LAT condensation process, $k_c(t)$ is related to the experimentally measured success probability, $P_{LAT}(t)$ (from Fig. 3d), by: $k_c(t) = \frac{\dot{P}_{LAT}(t)}{(1 - P_{LAT}(t))}$. In order to compare various simple kinetic proofreading schemes to the measured data, we consider a kinetic proofreading process consisting of $N$ steps, where each step (indexed by $i$) is a first order process with corresponding rate $\lambda_i$. The distribution of delay times between initiation of the process and final activation, here referred to as $f_D(t)$, is given by the successive convolution of the delay time distribution at each step: $f_D(t) = f_1(t) \otimes f_2(t) \otimes \ldots \otimes f_N(t)$. Note that if we use the time of the associated LAT condensation as a measure of the activation event for each pMHC:TCR complex, then the probability distribution function of delay time from pMHC:TCR binding until LAT condensation (given by $\dot{P}_{LAT}(t)$) is equivalent to $f_D(t)$. In the simplifying case where each of the $N$ steps occurs with equivalent rate $\lambda$, $f_D(t)$ reduces to a gamma distribution: $f_D(t) = \frac{\lambda^N}{(N-1)!} t^{N-1} e^{-\lambda t}$. The corresponding propensity function for the simplified $N$-step kinetic proofreading scheme, $k_c^N(t)$, is then related to $f_D(t)$ by:

$$k_c^N(t) = \frac{f_D(t)}{1 - \int_0^t f_D(t') dt'} \tag{9}$$

In the case where $f_D(t)$ is the gamma distribution derived above, there is an analytic form for $k_c(t)$:

$$k_c^\gamma(t) = \frac{\lambda^N t^{N-1} e^{-\lambda t}}{\Gamma(N) - \Gamma_N(\lambda t)} \tag{10}$$

Where $\Gamma(x)$ is the Gamma function, and $\Gamma_N(\lambda t)$ is the lower incomplete Gamma function, $\Gamma_N(\lambda t) = \int_0^{\lambda t} x^{N-1} e^{-x} dx$. The analytic expression $k_c^\gamma(t)$ was fit to the first 13 s. of the empirical $k_c(t)$ to generate the simple kinetic proofreading success rate curves seen in Fig. 3e.

## Estimation of Grb2 detection limit

To analyze the sensitivity of the imaging method used to track Grb2, the spots detected by TrackMate were further analyzed. Dense features of Grb2 are still observable at the membrane (Supplementary Fig. 4g) even if they don't always correlate with pMHC:TCR. For the algorithms used in this paper, the 5th percentile of spot detections (which approaches the limit of detection) had an $SNR = 1.4$. We use a definition of SNR as $\frac{I_{in} - I_{out}}{I_{out}}$, where $I_{in}$ is the mean intensity inside, $I_{out}$ is the mean intensity outside (in an annulus from the spot radius to 2x the spot radius). Low-expressing cells were used to create an intensity distribution for single molecule mNG-Grb2 at the membrane. The diffusion coefficient of free LAT-GFP was measured to be $2.8 \,\mu m^2/s$, suggesting that in 100 ms (the imaging exposure) the average displacement would be $1.05 \,\mu m$. Using a one square micron sampling window, we extrapolated the single-molecular brightness to the total intensity of one square micron of background for the live-cells expressing the LAT-Grb2 p2a construct (see Quantitative Fluorescence section below). This led to an estimated 20 molecules $\mu m^{-2}$ worth of fluorescence (from various diffusive species) of mNG-Grb2 present in

the background for these cells. The spot detection areas are ≈0.25 μm², suggesting that over a TCR, the detectable spatial window contains a steady state of 5 background "equivalents" of fluorescent mNG-Grb2 during camera exposure, and within this window a spot detection would indicate ≈7 more Grb2 molecules are present at the TCR over background (using SNR = 1.4). A failure to detect Grb2 over TCR indicates that fewer than 12 mNG-Grb2 are present in this area. Since the same MSCV promoter was used as for LAT (Supplementary Fig. 2b), an estimated 1:1 endogenous/exogenous ratio of Grb2 would indicate that fewer than ≈20 total Grb2 cannot be reliably detected at the TCR with this method.

## Quantitative fluorescence

To estimate the number of LAT-eGFP molecules within a LAT condensate we first calculate the number of fluorescent LAT molecules, $N_{LAT}^{Fl}$, by determining the mean integrated intensity of *single* LAT-eGFP molecules on the plasma membrane ($S_I$). This can be done either by finding a low-expressing cell or bleaching the cell down to single molecule densities and ensuring to analyze those particles with single step photobleaching. Ideally, we could divide the cluster intensity ($C_I$) by the mean PSF intensity:

$$N_{LAT}^{Fl} = \frac{C_I}{S_I} \qquad (11)$$

However, there are a few caveats. First, it is often the case that $S_I$ is imaged at different power, exposure, and EMCCD gain conditions than for bulk clusters $C_I$. This is done to stay within the dynamic range of the camera at both expression levels.

Fortunately, for many imaging systems, linear changes in power, exposure, and gain result in linear changes in intensity, but this must be verified. For example, the Andor iXon887 RealGain technology allows linear inference in the number of incident photoelectrons from fluorescent photons. And LED lasers allow for precise control of power input.

Once linearity in power, gain, and exposure is established then the following normalization will account for the shifting intensity values recorded under different conditions:

$$N_{LAT}^{Fl} = \frac{C_I / C_{gain} C_{power} C_{exposure}}{S_I / S_{gain} S_{power} S_{exposure}} \qquad (12)$$

For example, a given cluster intensity $C_I$ recorded at $C_{power} = 2$ will represent half the number of molecules if the same intensity value is recorded with $C_{power} = 4$.

The second caveat is to consider how much of the cellular background LAT is participating within the cluster. If there exists an equal density of free LAT within the cluster region as there does outside the cluster region, then one should use the net cluster intensity to determine the number of extra molecules that exist within the cluster region. However, if the density of free LAT within the cluster region is zero, then the total intensity of the cluster should be used to calculate the number of clustered molecules.

In practice, we used the midpoint between the net intensity and the total intensity: $C_I = \frac{1}{2}(C_{total} + C_{net})$.

Next, quantification of the western blot revealed the ratio of exogenous to endogenous LAT (Supplementary Fig. 2b), $\alpha = \frac{endo}{exo}$. Cell sorting showed that ~15% of T cells were transduced with GFP in the batch used for calibration, this percentage is crucial for correctly calculating $\alpha$ from the western blot. In this study, $\alpha$ was determined to be 0.60, and the resulting expression for total LAT is: $N_{LAT} = (1 + \alpha)N_{LAT}^{Fl}$.

## Reporting summary

Further information on research design is available in the Nature Portfolio Reporting Summary linked to this article.

## Data availability

The raw microscopy data and custom code used for the analysis can be downloaded from the following link (≈ 21 Gb): https://doi.org/10.5281/zenodo.7199954. A reporting summary for this Article is available as a Supplementary Information file. Source data are provided with this paper.

## Code availability

The custom code used to analyze the microscopy images is located at the same link as indicated in the "Data availability" section. See the readme file in the top folder of the download for setup and runtime information.

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

## Acknowledgements

Supported by NIH grant P01 AI091580 and by the Novo Nordisk Foundation under the Center for Geometrically Engineered Cellular Systems (NNF17OC0028176). We thank S. Hansen (University of Oregon) and S. Alvarez (Inscripta) for their contributions to the synthesis of p2a plasmids and cloning advice used in this study; A. Weiss (UCSF) for gifting a sample of Jurkat cells; F. Marangoni (Harvard Medical School) for providing the NFAT reporter plasmid; L. Teyton (Scripps Research Institute) and M. Davis (Stanford University) for providing the MHC and ICAM-1 bacmids; and Kole T. Roybal for the primary human T cells. The financial support for this work was provided by National Institutes of Health grant P01 AI091580 and by the Novo Nordisk Foundation Challenge Programme as part of the Center for Geometrically Engineered Cellular Systems.

## Author contributions

Conceptualization: J.T.G. and D.B.M.; methodology: D.B.M.; formal analysis: D.B.M.; investigation: D.B.M., M.K.O., S.T.L.-N., J.J.L., K.B.W., and S.M; resources: D.B.M, J.J.L., and S.K.; data curation: D.B.M., M.K.O., S.T.L.-N., and J.J.L.; writing—original draft: D.B.M. and J.T.G; writing—review and editing: D.B.M., M.K.O., S.T.L.-N., J.J.L., K.B.W., S.K, S.M., and J.T.G.; supervision: J.T.G.; funding acquisition: J.T.G.

## Competing interests

The authors declare no competing interests.
