## [Peer Review File · Nature Communications]

Discrete LAT condensates encode antigen information from
single pMHC:TCR binding eventsREVIEWER COMMENTS

Reviewer #1 T cell activation (Remarks to the Author):

The present study analysis at low pMHC densities (0.1–0.4 molecules μm^{-2}), how the information triggered by individual pMHC:TCR binding events translate into downstream signaling events. It uses a hybrid live cell-supported membrane interface functionalized with pMHC complexes of different dwell times and it tracks the formation, duration, and movement of individual pMHC:TCR binary complexes while simultaneously monitoring LAT condensation and NFAT nuclear translocation in response to each pMHC:TCR binding event. It revealed that a single long dwelling pMHC:TCR binding event sufficed to trigger formation of a LAT condensate containing a few hundred of LAT molecules and endowed with a mean lifetime of a few tens of seconds. Those LAT condensates began forming within 50-80 nm of the productive TCR engagement and some drifted a small distance away from the receptor at later time points. Therefore, LAT phosphorylation and condensation occur in contiguity to single pMHC:TCR binding events.

LAT condensation was also found to occur abruptly, after an extended delay from the originating pMHC:TCR binding event. Sustained phosphorylation LAT (as indirectly reported using a Grb2 reporter) remains undetectable above background prior to condensation. Therefore the delay between pMHC:TCR binding and LAT condensation suggest that it does not correspond to the time it takes for period of ZAP-70 kinase activity to build up a sufficient density of phosphorylated LAT before the phase transition occurs. Moreover, the resulting LAT condensates are self-limiting, and neither their size nor their lifetime was correlated with the duration of the originating pMHC:TCR binding event. Only the probability of forming a LAT condensate (per unit of time) is related to the pMHC:TCR binding dwell time. Interestingly, a LAT mutation that enhances the kinetics of ZAP-70 phosphorylation of residue Y136 of LAT and thereby boosts PLC- γ 1 recruitment decreased the delay time to LAT condensation. In the case of the MCC and T102S pMHC, it was also observed that distinct binding events, that are locally coincident, contribute to LAT condensation consistent with the previous author observation that short pMHC-TCR binding events that were spatially correlated and temporally sequential led to cellular activation.

Finally, the authors simultaneously monitored pMHC:TCR binding, LAT condensation, and NFAT translocation. For these experiments, T cells were incubated on bilayers with either MCC pMHC or T102S pMHC and monitored for 12 minutes. Even though a smaller proportion of cells activated in response to T102S pMHC, those activated T cells which had a comparable number of LAT condensates as T cells activating in response to MCC pMHC. Comparison of the NFAT translocation rates between T cells expressing LAT WT or LAT G135D further revealed that the presence of LAT G135D modestly increased NFAT translocation rates, suggesting that alterations in LAT condensation kinetics change the T cell responsiveness to shorter dwelling pMHC ligands. This comprehensive and elegant study further our understanding on how the information triggered by individual pMHC:TCR binding events translate into vital T cell signaling events. I do not have any major comments.

Minor comments

1/ Single molecule measurements of the pMHC:TCR:ZAP70 complex formation is used as a proxy of TCR triggering. It is an important assumption since it is used to conclude that the LAT condensates dissipate prior to the pMHC:TCR complex dissociation and that once formed all condensates function independently from the originating binding event. Is there any way of probing via an intracellular scFV-based probe a tyrosine residue of ZAP-70 linked to its enzymatic activity?

2/ It might be worth mentioning a recent study (doi.org/10.1084/jem.20211295) in which T cells were artificially cross-linked with TCR with specific antibodies and the dynamics of LAT signalosome formation monitored by quantitative interactomics. This last study markedly differs from the above experiments, which examine at single-cell level resolution how sparsely distributed pMHC:TCR binding events trigger individual LAT condensates. Intriguingly, these two orthogonal approaches concur to show that LAT condensates have a mean lifetime of a few tens of seconds that as shown in doi.org/10.1084/jem.20211295 results in part from the action of members of the 14-3-3 phosphoserine-phosphothreonine-binding protein family.

Reviewer #2 biostats, mathematical immunology (Remarks to the Author):

McAfee and colleagues have performed an extensive analysis of LAT condensation upon stimulation of T cells with various peptide-MHC complexes at various concentrations. The analysis, based on single-molecule imaging techniques applied to pMHC:TCR binding events and LAT condensates, provides rich data on the processes leading to T cell activation. These data will help to explain phenomena that have previously been observed in populations of T cells and are likely to be of substantial interest to researchers with an interest in understanding the function of the immune system. For example, the results shed light on the differing dynamics of immune responses at the single cell level between strongly bound antigens, compared to more weakly bound antigens that are present at higher concentrations. The paper includes estimates of several parameters that are likely to be important for models of T cell activation, including delay times between binding and LAT condensation and also explores the impact of LAT mutations on these delay parameters. The paper is very well written and clear and I have no major recommendations for improvement.

Minor points

- How were the bin boundaries determined (as shown for example in Fig. 3A)?
- Typo in Fig 3B legend: "The number of productive dwell time segments (red) for a particular time bin was the number binding events..." (missing 'of' – also missed in description of Fig3C).
- Could Fig 3E legend be rephrased? It was not clear in its current form, at least to me (should it state that gamma distributions were fitted to the effective nucleation rates?). Also there is a grammatical concordance error in the current phrasing: "The effective nucleation rate for various gamma distributions were fit" Should either be "was fit" or "effective nucleation rates"
- Line 281: Would be good to refer to the figure that supports this statement (Fig 3D lower panel?). It might make it easier to follow the statement if there was a horizontal line indicating this maximum probability in the plot (though this could also add clutter – the authors can decide if this is helpful or not).
- I could not follow this statement: "Note that if we define the associated LAT condensation as the full activation event for a TCR, then $f_D(t)$ is the derivative of the measured single TCR antigen discrimination function, $f_D(t) = d/dt P_{LAT}(t)$." Is it possible to make clearer why this is the case?
- L340: It's not really clear what's meant by "with the rise and fall shape of a gamma distribution". Is this a comment on the shape of the histogram? The histogram shape depends on the bin size. The gamma family of distributions is very flexible and can fit many distribution shapes. What is the significance of this histogram being similar in shape to a gamma distribution (and is that demonstrated?). It may be better to remove this comment unless it is informative in some way.
- "Data expressed simply as $x \pm y$, lacking SD or SE, arise from fit parameters, the error of fit was calculated from the square root of the covariance matrix when using the `curve_fit` method in the `scipy` python library" In addition to saying how these error values were calculated, could the authors explain what they are? I.e. what is the covariance matrix referred to here and how should its square root be interpreted?

Reviewer #3 imaging / T cell activation (Remarks to the Author):

In this paper McAfee et al. track individual pMHC binding events and investigate associated LAT condensates, using TIRF microscopy together with primary mouse CD4 AND TCR T cells and supported lipid bilayers decorated with ICAM-1 and MHC with MCC peptide, or altered peptide ligands with different affinities. They find that individual TCR-pMHC binding events are sufficient to trigger LAT condensates in their vicinity, but that neither the size of the LAT condensate, nor its duration, is correlated with the dwell time of the TCR-pMHC complex. Instead, TCR-pMHC dwell time increases the likelihood of condensate formation up to a maximum likelihood of 0.25 for each

individual binding event. They also find that the instantaneous likelihood of triggering a LAT condensate is maximal at ~15 seconds post TCR engagement, after which the likelihood decreases steadily to 0, which strongly suggests a negative feedback mechanism exists that dominates on longer timescales. LAT condenses rapidly upon nucleation and dissolves over time. Furthermore, LAT condensates seem to occur simultaneously with Grb2 enrichment, which the authors state as being strong evidence for a lack of slow local phosphorylated LAT buildup prior to condensate formation. Interestingly a G135D mutant of LAT which has accelerated ZAP70-dependent phosphorylation at the PLCgamma binding site, decreases the average delay between TCR-pMHC engagement and LAT condensate formation.

Overall I find this a very informative report that fills an important gap in our knowledge about LAT condensate formation. The authors highlight that the importance of observing consequences of single TCR-pMHC interaction in this context and I strongly agree with this. My principle concern is in the strength of two of their assertions, which I feel are inadequately supported by the presented data. My comments and questions are outlined below:

Main points:

1. I am not convinced by the assertion that there is not a buildup of phosphorylated LAT in the vicinity of TCR-pMHC interactions prior to LAT condensation. The evidence quoted relies heavily on the interpretation of the Grb2 imaging in figure 4. Interaction affinity of the Grb2 SH2 domain with pLAT is low (by the author's own measurements in another paper: mean dwell time of ~0.65 seconds, <https://www.pnas.org/doi/full/10.1073/pnas.1602602113>), and there is competing endogenous Grb2, so it is unlikely you will see significant accumulation of Grb2 outside of LAT condensates, where multivalent interactions with SOS1 and LAT will extend the effective dwell time. If more evidence cannot be produced to support this conclusion it would be better to soften the statement in lines 370-372 to account for this, something like "These results suggest that there is not an extended build up of pLAT in the vicinity of TCR prior to LAT condensate formation." Other statements in the conclusion should also be softened, eg lines 492-493 and lines 504-505.

2. The other conclusion that is strongly stated but only loosely supported by the data is the concept that multiple binding events in the same vicinity can have an additive effect on the probability of LAT condensation. Only the one example from one condensation event in figure 5B is shown to support this. It would be more compelling if the authors could provide quantitative data, for example by extending their analysis of the probability of LAT condensate formation to the "effective dwell time" events suggested in Fig 5B left panel. Also, if locally correlated binding events at higher ligand densities increase the probability of LAT condensates this suggests that there is some local "memory", which erodes proofreading and this should also be mentioned. Furthermore this local "memory" is most easily explained by an enhanced local concentration of pLAT that dissipates relatively slowly, which also relates to my first main point.

Smaller points:

1. Does the G135D mutant change the overall likelihood of condensate formation above the 0.25 limit for the wild-type? This would give more information about the competition of rates between condensate formation and any negative feedback that limits their formation at later timepoints.

2. Lines 318-320: This is a key assumption of their model of kinetic proofreading. It seems reasonable that this might be the case, but it is not necessarily true that LAT condensation is the full activation event for the TCR. Some acknowledgment that this the assumption underpinning the conclusions made would aid in transparency readers unfamiliar with the kinetic proofreading literature.

3. Lines 37-38: This recent paper should also be quoted <https://doi.org/10.7554/eLife.67092>

Response to the reviewers' comments

Reference numbers in the responses below refer to the main text references for the revised manuscript.

Reviewer #1

The present study analysis at low pMHC densities (0.1–0.4 molecules μm^{-2}), how the information triggered by individual pMHC:TCR binding events translate into downstream signaling events. It uses a hybrid live cell-supported membrane interface functionalized with pMHC complexes of different dwell times and it tracks the formation, duration, and movement of individual pMHC:TCR binary complexes while simultaneously monitoring LAT condensation and NFAT nuclear translocation in response to each pMHC:TCR binding event. It revealed that a single long dwelling pMHC:TCR binding event sufficed to trigger formation of a LAT condensate containing a few hundred of LAT molecules and endowed with a mean lifetime of a few tens of seconds. Those LAT condensates began forming within 50-80 nm of the productive TCR engagement and some drifted a small distance away from the receptor at later time points. Therefore, LAT phosphorylation and condensation occur in contiguity to single pMHC:TCR binding events.

LAT condensation was also found to occur abruptly, after an extended delay from the originating pMHC:TCR binding event. Sustained phosphorylation LAT (as indirectly reported using a Grb2 reporter) remains undetectable above background prior to condensation. Therefore the delay between pMHC:TCR binding and LAT condensation suggest that it does not correspond to the time it takes for period of ZAP-70 kinase activity to build up a sufficient density of phosphorylated LAT before the phase transition occurs. Moreover, the resulting LAT condensates are self-limiting, and neither their size nor their lifetime was correlated with the duration of the originating pMHC:TCR binding event. Only the probability of forming a LAT condensate (per unit of time) is related to the pMHC:TCR binding dwell time. Interestingly, a LAT mutation that enhances the kinetics of ZAP-70 phosphorylation of residue Y136 of LAT and thereby boosts PLC- γ 1 recruitment decreased the delay time to LAT condensation. In the case of the MCC and T102S pMHC, it was also observed that distinct binding events, that are locally coincident, contribute to LAT condensation consistent with the previous author observation that short pMHC-TCR binding events that were spatially correlated and temporally sequential led to cellular activation.

Finally, the authors simultaneously monitored pMHC:TCR binding, LAT condensation, and NFAT translocation. For these experiments, T cells were incubated on bilayers with either MCC pMHC or T102S pMHC and monitored for 12 minutes. Even though a smaller proportion of cells activated in response to T102S pMHC, those activated T cells which had a comparable number of LAT condensates as T cells activating in response to MCC pMHC. Comparison of the NFAT translocation rates between T cells expressing LAT WT or LAT G135D further revealed that the presence of LAT G135D modestly

increased NFAT translocation rates, suggesting that alterations in LAT condensation kinetics change the T cell responsiveness to shorter dwelling pMHC ligands. This comprehensive and elegant study further our understanding on how the information triggered by individual pMHC:TCR binding events translate into vital T cell signaling events. I do not have any major comments.

Minor comments

1/ Single molecule measurements of the pMHC:TCR:ZAP70 complex formation is used as a proxy of TCR triggering. It is an important assumption since it is used to conclude that the LAT condensates dissipate prior to the pMHC:TCR complex dissociation and that once formed all condensates function independently from the originating binding event. Is there any way of probing via an intracellular scFV-based probe a tyrosine residue of ZAP-70 linked to its enzymatic activity?

2/ It might be worth mentioning a recent study (doi.org/10.1084/jem.20211295) in which T cells were artificially cross-linked with TCR with specific antibodies and the dynamics of LAT signalosome formation monitored by quantitative interactomics. This last study markedly differs from the above experiments, which examine at single-cell level resolution how sparsely distributed pMHC:TCR binding events trigger individual LAT condensates. Intriguingly, these two orthogonal approaches concur to show that LAT condensates have a mean lifetime of a few tens of seconds that as shown in doi.org/10.1084/jem.20211295 results in part from the action of members of the 14-3-3 phosphoserine-phosphothreonine-binding protein family.

Point-by-point response to Reviewer #1

Single molecule measurements of the pMHC:TCR:ZAP70 complex formation is used as a proxy of TCR triggering.

Response: First, we would like to clarify that our measurements of pMHC:TCR binding are not used as a surrogate observation of TCR triggering. To the contrary, these are used only (and directly) to identify pMHC:TCR complexes. Evidence of successful (or failed) downstream signaling from each pMHC:TCR complex is independently observed by monitoring co-localized LAT condensation, which is a demonstrable result of successful downstream signaling from the pMHC:TCR complex, including ZAP70 kinase activation. For the observed pMHC:TCR complexes that fail to trigger LAT condensation, we cannot say with certainty if ZAP70 was or was not activated based on our measurements alone. We only know that downstream signaling in the form of LAT condensation is evidently not achieved for these particular binding events. However, a large body of literature indicates that the timescale of ZAP70 activation must occur within a few seconds of pMHC:TCR engagement, which is much shorter than the mean delay time to LAT condensation. From this it seems most likely that ZAP70 has been activated, but that LAT condensation itself presents another threshold that is not necessarily crossed. This, however, is not a conclusion of the present manuscript. Indeed, subsequent work (well beyond the scope of the present paper) is examining the nature of the LAT condensation phase transition and how it provides a thresholding event capable of blocking further signaling from weak ZAP70 kinase activity.

It [pMHC:TCR:ZAP70 complex formation] is an important assumption since it is used to conclude that the LAT condensates dissipate prior to the pMHC:TCR complex dissociation and that once formed all condensates function independently from the originating binding event.

Response: Our conclusion that LAT condensates can dissipate prior to pMHC:TCR unbinding is based on direct observations in a subset of the pMHC:TCR binding events (e.g. Fig. 1G). The statement that condensates generally function independently from one another is based on the observation that there were no obvious correlations between the properties of LAT condensates (e.g. the presence or absence of a nearby condensate produced no noticeable effect). There is ultimately integration of the signal from multiple LAT condensates to contribute to the overall cellular decision to activate (e.g. as discussed in the context of Fig. 6).

Is there any way of probing via an intracellular scFV-based probe a tyrosine residue of ZAP-70 linked to its enzymatic activity?

Response: The experiment suggested by the referee to use an intracellular scFV-based probe to monitor ZAP70 activity state is interesting. However, we feel this would go significantly beyond the scope of the present manuscript for several reasons: First: This manuscript does not study ZAP70 per se. Rather, the experimental approach remains entirely agnostic to the activation timescale of ZAP and focuses on the relation between pMHC:TCR binding and co-localized LAT condensation, both of which are directly observed without 3rd-party probes. Second: The svFV-probe experiment introduces additional complexities, especially from the probe kinetics themselves. We are monitoring signaling reaction steps at the single molecule level with sub-second time resolution. It is quite unclear if svFV probes would detect ZAP activation events with such high time resolution—so an entirely separate set of experiments to validate probe kinetics etc. would be necessary to bring this type of measurement to the same resolution as other measurements in the paper. We certainly agree with the referee that this would be a good experiment for another study focused on ZAP activation kinetics.

It might be worth mentioning a recent study (doi.org/10.1084/jem.20211295) in which T cells were artificially cross-linked with TCR with specific antibodies and the dynamics of LAT signalosome formation monitored by quantitative interactomics. This last study markedly differs from the above experiments, which examine at single-cell level resolution how sparsely distributed pMHC:TCR binding events trigger individual LAT condensates. Intriguingly, these two orthogonal approaches concur to show that LAT condensates have a mean lifetime of a few tens of seconds that as shown in doi.org/10.1084/jem.20211295 results in part from the action of members of the 14-3-3 phosphoserine-phosphothreonine-binding protein family.

Response: We thank the reviewer for highlighting a recent study that corroborates an observation in our manuscript and have included this reference in our revision. LAT lifetimes are now described in the first result section, with the following:

L172 (new): The LAT condensates are self-limiting with a mean lifetime of $\langle \tau_{condensate} \rangle \approx 30$ s.; a similar limiting lifetime for LAT condensates has recently been estimated indirectly from mass spectrometry studies⁷³ on cell populations.

Reviewer #2

McAfee and colleagues have performed an extensive analysis of LAT condensation upon stimulation of T cells with various peptide-MHC complexes at various concentrations. The analysis, based on single-molecule imaging techniques applied to pMHC:TCR binding events and LAT condensates, provides rich data on the processes leading to T cell activation. These data will help to explain phenomena that have previously been observed in populations of T cells and are likely to be of substantial to researchers with an interest in understanding the function of the immune system. For example, the results shed light on the differing dynamics of immune responses at the single cell level between strongly bound antigens, compared to more weakly bound antigens that are present at higher concentrations. The paper includes estimates of several parameters that are likely to be important for models of T cell activation, including delay times between binding and LAT condensation and also explores the impact of LAT mutations on these delay parameters. The paper is very well written and clear and I have no major recommendations for improvement.

Minor points

- How were the bin boundaries determined (as shown for example in Fig. 3A)?
- Typo in Fig 3B legend: "The number of productive dwell time segments (red) for a particular time bin was the number binding events..." (missing 'of' – also missed in description of Fig3C).
- Could Fig 3E legend be rephrased? It was not clear in its current form, at least to me (should it state that gamma distributions were fitted to the effective nucleation rates?). Also there is a grammatical concordance error in the current phrasing: "The effective nucleation rate for various gamma distributions were fit" Should either be "was fit" or "effective nucleation rates"
- Line 281: Would be good to refer to the figure that supports this statement (Fig 3D lower panel?). It might make it easier to follow the statement if there was a horizontal line indicating this maximum probability in the plot (though this could also add clutter – the authors can decide if this is helpful or not).
- I could not follow this statement: "Note that if we define the associated LAT condensation as the full activation event for a TCR, then $f_D(t)$ is the derivative of the measured single TCR antigen discrimination function, $f_D(t) = d/dt P_{LAT}(t)$." Is it possible to make clearer why this is the case?
- L340: It's not really clear what's meant by "with the rise and fall shape of a gamma distribution". Is this a comment on the shape of the histogram? The histogram shape depends on the bin size. The gamma family of distributions is very flexible and can fit many distribution shapes. What is the significance of this histogram being similar in shape to a gamma distribution (and is that demonstrated?). It may be better to remove this comment unless it is informative in some way.

- “Data expressed simply as $x \pm y$, lacking SD or SE, arise from fit parameters, the error of fit was calculated from the square root of the covariance matrix when using the `curve_fit` method in the `scipy` python library” In addition to saying how these error values were calculated, could the authors explain what they are? I.e. what is the covariance matrix referred to here and how should its square root be interpreted?

Point-by-point response to Reviewer #2

- How were the bin boundaries determined (as shown for example in Fig. 3A)?

Response: The following was added in the figure legend for Fig. 3:

Linearly increasing bin widths were used to improve the sampling rate of rare long-binding events.

- Typo in Fig 3B legend: “The number of productive dwell time segments (red) for a particular time bin was the number binding events...” (missing ‘of’ – also missed in description of Fig3C).

Response: We have now fixed these typos.

- Could Fig 3E legend be rephrased? It was not clear in its current form, at least to me (should it state that gamma distributions were fitted to the effective nucleation rates?). Also there is a grammatical concordance error in the current phrasing: “The effective nucleation rate for various gamma distributions were fit” Should either be “was fit” or “effective nucleation rates”

Response: 3E (legend) has been changed to the following for clarity:

Nucleation rates for various hypothetical simple kinetic proofreading schemes (N=1, 2, or 3 steps) are plotted to illustrate key differences compared with the observed effective nucleation rate (see “ $k_c(t)$ as a Propensity Function” in Methods for more details).

- Line 281: Would be good to refer to the figure that supports this statement (Fig 3D lower panel?). It might make it easier to follow the statement if there was a horizontal line indicating this maximum probability in the plot (though this could also add clutter – the authors can decide if this is helpful or not).

Response: A reference to 3D was added.

- I could not follow this statement: “Note that if we define the associated LAT condensation as the full activation event for a TCR, then $f_D(t)$ is the derivative of the measured single TCR antigen discrimination function, $f_D(t) = d/dt P_{LAT}(t)$.” Is it possible to make clearer why this is the case?

Response: To clarify L318, we have moved the detailed discussion to a new section in Methods and made the following changes to that paragraph, which now reads:

L270(new): Kinetic proofreading processes are often examined in terms of a steady state rate of activation of a downstream signaling event^{8,19,78}. A more complete description of a kinetic proofreading mechanism is provided by the delay time distribution between initial ligand engagement of the receptor and the subsequent activation event, from which the propensity function ($k_c(t)$) for successful activation can be determined (see $k_c(t)$ as a *Propensity Function* in Methods).

- L340: It's not really clear what's meant by "with the rise and fall shape of a gamma distribution". Is this a comment on the shape of the histogram? The histogram shape depends on the bin size. The gamma family of distributions is very flexible and can fit many distribution shapes. What is the significance of this histogram being similar in shape to a gamma distribution (and is that demonstrated?). It may be better to remove this comment unless it is informative in some way.

Response: we agree that the gamma comparison is qualitative and not quantitative and has been removed to avoid confusion.

- "Data expressed simply as $x \pm y$, lacking SD or SE, arise from fit parameters, the error of fit was calculated from the square root of the covariance matrix when using the curve_fit method in the scipy python library" In addition to saying how these error values were calculated, could the authors explain what they are? I.e. what is the covariance matrix referred to here and how should its square root be interpreted?

Response: To clarify the SD/SE we have added the following:

L628(new): The `curve_fit` method from the `scipy` library returns a covariance matrix `pcov`, that is based on scaling the error (sigma) by a constant factor. This constant is set by demanding that the reduced `chisq` for the optimal parameters, `popt`, when using the scaled sigma equals unity. In other words, `sigma` is scaled to match the sample variance of the residuals after the fit. See https://docs.scipy.org/doc/scipy/reference/generated/scipy.optimize.curve_fit.html for more information.

Reviewer #3

In this paper McAfee et al. track individual pMHC binding events and investigate associated LAT condensates, using TIRF microscopy together with primary mouse CD4 AND TCR T cells and supported lipid bilayers decorated with ICAM-1 and MHC with MCC peptide, or altered peptide ligands with

different affinities. They find that individual TCR-pMHC binding events are sufficient to trigger LAT condensates in their vicinity, but that neither the size of the LAT condensate, nor its duration, is correlated with the dwell time of the TCR-pMHC complex. Instead, TCR-pMHC dwell time increases the likelihood of condensate formation up to a maximum likelihood of 0.25 for each individual binding event. They also find that the instantaneous likelihood of triggering a LAT condensate is maximal at ~15 seconds post TCR engagement, after which the likelihood decreases steadily to 0, which strongly suggests a negative feedback mechanism exists that dominates on longer timescales. LAT condenses rapidly

upon nucleation and dissolves over time. Furthermore, LAT condensates seem to occur simultaneously with Grb2 enrichment, which the authors state as being strong evidence for a lack of slow local phosphorylated LAT buildup prior to condensate formation. Interestingly a G135D mutant of LAT which has accelerated ZAP70-dependent phosphorylation at the PLCgamma binding site, decreases the average delay between TCR-pMHC engagement and LAT condensate formation.

Overall I find this a very informative report that fills an important gap in our knowledge about LAT condensate formation. The authors highlight that the importance of observing consequences of single TCR-pMHC interaction in this context and I strongly agree with this. My principle concern is in the strength of two of their assertions, which I feel are inadequately supported by the presented data. My comments and questions are outlined below:

Main points:

1. I am not convinced by the assertion that there is not a buildup of phosphorylated LAT in the vicinity of TCR-pMHC interactions prior to LAT condensation. The evidence quoted relies heavily on the interpretation of the Grb2 imaging in figure 4. Interaction affinity of the Grb2 SH2 domain with pLAT is low (by the author's own measurements in another paper: mean dwell time of ~0.65 seconds, <https://www.pnas.org/doi/full/10.1073/pnas.1602602113>), and there is competing endogenous Grb2, so it is unlikely you will see significant accumulation of Grb2 outside of LAT condensates, where multivalent interactions with SOS1 and LAT will extend the effective dwell time. If more evidence cannot be produced to support this conclusion it would be better to soften the statement in lines 370-372 to account for this, something like "These results suggest that there is not an extended build up of pLAT in the vicinity of TCR prior to LAT condensate formation." Other statements in the conclusion should also be softened, eg lines 492-493 and lines 504-505.

2. The other conclusion that is strongly stated but only loosely supported by the data is the concept that multiple binding events in the same vicinity can have an additive effect on the probability of LAT condensation. Only the one example from one condensation event in figure 5B is shown to support this. It would be more compelling if the authors could provide quantitative data, for example by extending their analysis of the probability of LAT condensate formation to the "effective dwell time" events suggested in Fig 5B left panel. Also, if locally correlated binding events at higher ligand densities increase the probability of LAT condensates this suggests that there is some local "memory", which erodes proofreading and this should also be mentioned. Furthermore this local "memory" is

most easily explained by an enhanced local concentration of pLAT that dissipates relatively slowly, which also relates to my first main point.

Smaller points:

1. Does the G135D mutant change the overall likelihood of condensate formation above the 0.25 limit for the wild-type? This would give more information about the competition of rates between condensate formation and any negative feedback that limits their formation at later timepoints.
2. Lines 318-320: This is a key assumption of their model of kinetic proofreading. It seems reasonable that this might be the case, but it is not necessarily true that LAT condensation is the full activation event for the TCR. Some acknowledgment that this the assumption underpinning the conclusions made would aid in transparency readers unfamiliar with the kinetic proofreading literature.
3. Lines 37-38: This recent paper should also be quoted <https://doi.org/10.7554/eLife.670921>. I am not convinced by the assertion that there is not a buildup of phosphorylated LAT in the vicinity of TCR-pMHC interactions prior to LAT condensation. The evidence quoted relies heavily on the interpretation of the Grb2 imaging in figure 4. Interaction affinity of the Grb2 SH2 domain with pLAT is low (by the author's own measurements in another paper: mean dwell time of ~0.65 seconds, <https://www.pnas.org/doi/full/10.1073/pnas.1602602113>), and there is competing endogenous Grb2, so it is unlikely you will see significant accumulation of Grb2 outside of LAT condensates, where multivalent interactions with SOS1 and LAT will extend the effective dwell time. If more evidence cannot be produced to support this conclusion it would be better to soften the statement in lines 370-372 to account for this, something like "These results suggest that there is not an extended buildup of pLAT in the vicinity of TCR prior to LAT condensate formation." Other statements in the conclusion should also be softened, eg lines 492-493 and lines 504-505.

Point-by-point response to Reviewer #3

Main point 1

1. I am not convinced by the assertion that there is not a buildup of phosphorylated LAT in the vicinity of TCR-pMHC interactions prior to LAT condensation. The evidence quoted relies heavily on the interpretation of the Grb2 imaging in figure 4. Interaction affinity of the Grb2 SH2 domain with pLAT is low (by the author's own measurements in another paper: mean dwell time of ~0.65 seconds, <https://www.pnas.org/doi/full/10.1073/pnas.1602602113>), and there is competing endogenous Grb2, so it is unlikely you will see significant accumulation of Grb2 outside of LAT condensates, where multivalent interactions with SOS1 and LAT will extend the effective dwell time. If more evidence cannot be produced to support this conclusion it would be better to soften the statement in lines 370-372 to account for this, something like "These results suggest that there is not an extended build up of pLAT in the vicinity of TCR prior to LAT condensate formation." Other statements in the conclusion should also be softened, eg lines 492-493 and lines 504-505.

Response: We thank the reviewer for their assessment of our manuscript. With regards to “main point #1”, in the lines mentioned, we have clarified that we are not arguing that no pLAT accumulates prior to LAT condensation, but instead, it is not detectable above background and there is a discontinuity in the rate of pLAT accumulation coupled to the LAT condensation event itself. This observation demonstrates that it is not sustained GRB2 accumulation that drives condensation, but rather a low-level density fluctuation of pLAT (and associated Grb2). We have contextualized these remarks with a new figure (Supplementary Fig. 4G) and an analysis of our limit of detection. In the main text we have added:

L317(new): Based on the detection limit in these imaging experiments, fewer than ≈ 20 Grb2 molecules are sustainably localized in the vicinity of the pMHC:TCR complex prior to nucleation of LAT condensation, though momentary fluctuations certainly occur (see Supplementary Fig. 4G and *Estimation of Grb2 detection limit* in Methods).

Main point 2

2. The other conclusion that is strongly stated but only loosely supported by the data is the concept that multiple binding events in the same vicinity can have an additive effect on the probability of LAT condensation. Only the one example from one condensation event in figure 5B is shown to support this. It would be more compelling if the authors could provide quantitative data, for example by extending their analysis of the probability of LAT condensate formation to the "effective dwell time" events suggested in Fig 5B left panel. Also, if locally correlated binding events at higher ligand densities increase the probability of LAT condensates this suggests that there is some local "memory", which erodes proofreading and this should also be mentioned. Furthermore, this local "memory" is most easily explained by an enhanced local concentration of pLAT that dissipates relatively slowly, which also relates to my first main point.

Response: We are completely in agreement with the points raised by the referee; the original text was apparently insufficiently clear on these points. First, the primary data supporting a role for coordinated short binding events is published in other work from our lab¹², where quantitative assessments along the lines suggested by the referee are included. That earlier work established the statistical significance of correlated events, but without LAT imaging. The only readout at that point was cellular activation observed via NFAT translocation.

We include the data in the original Fig. 5B (now Supplementary Figure 5C) to provide anecdotal evidence in the observation of a very difficult to capture correlated binding event. While this data itself is not quantifiable due to small numbers of observed events, we feel it is valuable to include in conjunction with the earlier published result. We have edited the text to more clearly describe this and the figure has been moved to Supplementary Materials, both to save space and since this is not a major conclusion of the work.

As far as the ‘local memory’—this is exactly the conclusion we intend to draw. There is a brief local memory that enables more than one pMHC:TCR to contribute to the successful production of a single LAT condensate. As the reviewer mentions, we also speculate that it is indeed brief localized accumulation of pLAT that is this memory. Obviously, these nuanced interpretations were not stated clearly in our original draft and we have carefully edited the main text as follows:

L322(new): This observation rules out one possible cause of the delay between pMHC:TCR binding and LAT condensation: that a high localized density of phosphorylated LAT must accumulate before the condensation phase transition can occur. Rather, it appears that a relatively low-density fluctuation in phosphorylated LAT from the competing kinase-phosphatase reactions themselves is the nucleating event. This is followed by a rapid accumulation of phosphorylated LAT (and associated Grb2) in the growing condensate, which is evidence for some form of positive feedback (such as phosphatase exclusion from the LAT condensate³⁹).

Smaller points

1. Does the G135D mutant change the overall likelihood of condensate formation above the 0.25 limit for the wild-type? This would give more information about the competition of rates between condensate formation and any negative feedback that limits their formation at later timepoints.

Response: This analysis was not performed due to technical limitations. Since pMHC is much more productive under LATG135D conditions, many more LAT condensates coexist simultaneously across the T cell interface (typically 5-10, as opposed to 2-4 under normal low-density pMHC conditions). While it remains possible to isolate individual binding events that were successful (which data leads to Fig. 4E-F); it becomes difficult to track “all” binding events, since many will cross pre-existing LAT condensates and contaminate the pMHC trajectory of having a “clean” history.

2. Lines 318-320: This is a key assumption of their model of kinetic proofreading. It seems reasonable that this might be the case, but it is not necessarily true that LAT condensation is the full activation event for the TCR. Some acknowledgment that this the assumption underpinning the conclusions made would aid in transparency readers unfamiliar with the kinetic proofreading literature.

Response: To clarify L318, we have moved the detailed discussion to a new section in Methods and made the following changes to that paragraph, which now reads:

L270(new): Kinetic proofreading processes are often examined in terms of a steady state rate of activation of a downstream signaling event^{8,19,78}. A more complete description of a kinetic proofreading mechanism is provided by the delay time distribution between initial ligand engagement of the receptor and the subsequent activation event, from which the propensity function ($k_c(t)$) for successful activation can be determined (see $k_c(t)$ as a *Propensity Function* in Methods).

3. Lines 37-38: This recent paper should also be quoted <https://doi.org/10.7554/eLife.67092>

We have cited the Pettmann paper (ref. 19) but will emphasize its relevance in the lines indicated at the reviewer’s suggestion.

REVIEWERS' COMMENTS

Reviewer #1 (Remarks to the Author):

None

Reviewer #2 (Remarks to the Author):

I raised only relatively minor points in my review and they have been addressed adequately by the authors.

Reviewer #3 (Remarks to the Author):

I thank the authors for addressing my comments. The rewording and additional information about local Grb2 and LAT accumulation before condensation has clarified the message and satisfied my concerns.